



# Trawling effects on biogeochemical processes are mediated by fauna in high energy biogenic reef–inhabited coastal sediments

Justin C. Tiano[1,2], Jochen Depestele[3], Gert Van Hoey[2,3], João Fernandes[4,5], Pieter van Rijswijk[1], Karline Soetaert[1,2]

[1]Royal Netherlands Institute for Sea Research (NIOZ) and Utrecht University, Department of 5 Estuarine and Delta Systems, PO box 140, 4400 AC, Yerseke, The Netherlands
[2]Ghent University, Department of Biology, Marine Biology Section, Krijgslaan 281/S8, 9000 Ghent, 9 Belgium
[3]Research Institute for Agriculture, Fisheries and Food (ILVO), Fisheries Research Group, Ankerstraat 1, 8400 Oostende, Belgium
[4]CETEMARES, Instituto Politécnico de Leiria, Avenida do Porto de Pesca 30, 2520-620 Peniche, Portugal
[5]Lab. MAREFOZ, Incubadora de Empresas da Figueira da Foz, R. Acácias lote 40A, 3090-380 Figueira da Foz, Portugal

*Correspondence to*: Justin C. Tiano (Justin.Tiano@nioz.nl)

**Abstract.** Dynamic, sandy environments are generally less vulnerable to mechanical stress compared to silty, low energy habitats. Biogenic reef communities, however, may provide an exception to this. This study explores the physical, biological and biogeochemical effects of electric pulse and tickler chain rigged beam trawls on a coastal ecosystem dominated by the tube building polychaete, *Lanice conchilega*. With a before-after-control-impact (BACI) in situ study, we detected a ~1 cm bathymetric deepening after trawling associated with significant losses in benthic chlorophyll *a* caused from both fishing gears. Trawled sediments exhibited physical mixing ($D_B$) values similar to areas with higher bioturbation. Tickler chain trawls significantly reduced sediment oxygen consumption (57%), total organic matter mineralization (56%), denitrification (61%), nitrification (60%), and total benthos densities (52%) while pulse trawls had no statistically significant impact on these parameters. Before trawling, significant relationships could be found between *L. conchilega* and very fine sand fractions, oxygen and nitrate fluxes, taxon densities and species richness, however, the trawl disturbances from both gears disrupted these connections. Our results suggest a reduced average effect for pulse compared to tickler chain beam trawls for several ecological and biogeochemical characteristics though their impact was still significant for *L. conchilega* and associated species. This study also suggests that faunal-mediated ecosystem functions in *L. conchilega* dominated habitats may be sensitive to relatively shallow sediment penetration from trawl gears and should be considered when assessing habitat vulnerability.

## 1 Introduction

Marine sediment ecosystems are important drivers for benthic pelagic coupling (Mermillod-Blondin and Rosenberg, 2006) and carbon mineralization (Seiter et al., 2005). Biogenic reef habitats in particular, such as those created by the tube-building polychaete, *Lanice conchilega*, are able to capture substantial amounts of organic carbon compared to adjacent sediments (De



Smet et al., 2016a) resulting in dense and biologically diverse benthic communities (Van Hoey et al., 2008). This species may be found in various North Sea habitats but generally prefer shallow muddy and fine sands where it may exhibit densities up to 1,000 individuals per m$^2$ (Van Hoey et al., 2008). Bottom trawling activities are a common occurrence in North Sea *L.*
*conchilega* dominated habitats (Rabaut, 2009) and present a considerable threat to biogenic reef ecosystems (Kaiser et al., 2002; Fariñas-Franco et al., 2018). Biological resilience to bottom trawl impacts have been linked to both *L. conchilega* reefs (Rabaut et al., 2008) and the dynamic sandy habitats they are often found in (Kaiser et al., 2006; Sciberras et al., 2016; Rijnsdorp et al., 2020a). Consequences of trawling on more holistic ecosystem processes such as benthic pelagic coupling, however, have not been thoroughly examined in these systems.

Bottom fishing affects carbon cycling by displacing bottom dwelling organisms (Hiddink et al., 2017; Sciberras et al., 2018), removing sediment structures needed to maximize biogeochemical processes (Depestele et al., 2018; Ferguson et al., 2020), and resuspending fresh organic material from the sediment surface, which can result in reduced mineralization of organic carbon (Tiano et., al. 2019; Paradis et al., 2020; Bradshaw et al., 2021; Morys et al., 2021). Conversely, trawling may also increase benthic carbon mineralization through depositional effects or if refractory organic fractions are mixed with and
reactivated by fresh organic matter (van de Velde et al., 2018). Trawl-induced mortality to benthic organisms can simplify benthic communities and can lead to a loss in functional diversity (Tillin et al., 2006; Tiano et al., 2020). The removal (significant or partial) of an abundant ecosystem engineer such as *L. conchilega* and associated fauna would likely have major impacts on biogeochemical and community dynamics (Rabaut et al., 2008; Van Hoey et al., 2008; De Smet et al., 2016a; Foshtomi et al., 2018).

The severity of bottom trawl impact depends on the size and type of fishing gear used and speed at which it drags over the seabed (Eigaard et al., 2016; Depestele et al., 2016; Depestele et al., 2018; Rijnsdorp et al,. 2020a, 2020b). In the southern North Sea, beam trawls rigged with tickler chains mechanically stimulate flatfish up from the seafloor, facilitating their capture. An alternative fishing gear, "pulse trawls", has been used to expose the seabed to electrical fields, causing the immobilization of target fish for oncoming bottom nets (Soetaert et al., 2015). Previous comparisons conducted in the Frisian
Front, a relatively homogenous offshore habitat known for its high abundances and diversity of benthic organisms (Dauwe et al., 1998; Dewick et al., 2002), have found a reduced impact of pulse compared to tickler rigged trawls on physical (Depestele et al., 2018) and biogeochemical parameters (Tiano et al., 2019) but impacts on benthic communities were similar for both gears (Tiano et al., 2020). Dynamic nearshore habitats, in contrast, may provide a more challenging environment for detecting significant anthropogenic alterations.

Benthic ecosystems characterized by coarse sediment and high levels of natural disturbance, typically display more resistance (sensitivity to direct impact) and resilience (recovery potential) to bottom fishing compared to muddy and more hydrodynamically static ecosystems (Kaiser et al., 2002; Bolam et al., 2014; Sciberras et al., 2016; Rijnsdorp et al., 2020a). Macrobenthos found in sandy, well-sorted sediments, tend to be less sensitive to trawl disturbance than in muddy poorly sorted habitats (Bolam et al., 2014; van Denderen et al., 2015). Communities dominated by three-dimensional biological features,





such biogenic reefs, seem to be an exception to this and can be vulnerable to trawl disturbance regardless of hydrodynamic energy level or substrate type (Kaiser et al., 2002; Grabowski et al., 2014).

The current study uses a combination of water column, sonar, sediment profile imagery, box core sampling, sediment incubation and porewater nutrient extraction techniques to provide a comprehensive assessment of bottom trawling on ecosystem functioning. We investigate the physical, biological and biogeochemical effects from two types of fishing gears in

a biogenic reef habitat. This integrated data collection allows us to explore the links between pure mechanical trawling effects, disturbance-induced changes to faunal-mediated biogeochemical functions and their collective impact on benthic pelagic coupling. Our main objective was to measure the acute ecosystem effects of electric pulse and tickler chain beam trawls in a high-energy coastal system using a before-after control-impact (BACI) design. Furthermore, the high abundances of *Lanice conchilega* in the region, allowed us to quantify the effect of this species on ecological and biogeochemical parameters, and

how this may change due to fishing disturbance.

## 2 Methods

### 2.1 Study area

This study took place in shallow subtidal waters (8 – 10 m depth), 10 km away from the Netherlands coast close to the Dutch and Belgian EEZ (exclusive economic zone) border (Fig. 1). This region, known as the "Vlakte van de Raan", is an area known

for its subtidal sandbanks, which create a mosaic of habitat areas of varying ecological value (Degraer and Hostens, 2016, Pecceu et al., 2016). The experimental areas were characterized by "fine sand" with an average median grain size of 200.5 µm and an average chlorophyll *a* content (0 – 10 cm depth) of 13.1 µg g$^{-1}$ (Fig. S1 in the Supplement). The exact location for this study within the Vlakte van de Raan (Fig. 1), was chosen due to its relatively low levels of bottom trawl disturbance (no flatfish or shrimp trawl disturbance in one year as verified with vessel monitoring data) and the high macrobenthic densities and

biomass (Craeymeersch et al., 2006).

### 2.2 Experimental trawling design

The effects of acute fishing disturbance from pulse and tickler chain rigged beam trawls were evaluated in controlled experimental settings following a beyond BACI protocol (Underwood, 1992). Three experimental treatments were conducted: (1) tickler chain beam trawling (hereafter referred to as "Tickler" treatment or tickler trawls), (2) pulse trawling with and (3)

without electricity turned on (hereafter referred to as "Pulse-ON" and "Pulse-OFF" treatment). Each treatment was repeated three times in spatially separated areas (Pulse-ON = A, B, C; Pulse-OFF = D, E, F, Tickler = G, H, I), resulting in a total of nine experimental plots of 50 by 300 m (15,000 m² per plot; Fig. 1). Individual plots were subdivided by three different sampling stations representing the western, middle, and eastern portions of the plot. Two additional untrawled reference stations (R1, R2) were chosen within the experimental area (Fig. 1).



Pulse trawls (Delmaco; Soetaert et al., 2015) were deployed from the commercial fishing vessel (FV) *TH6 Johanna Cornelia* (24 m) and tickler trawls from the FV *YE238 Geertruida* (22 m). The vessels were relatively small trawlers (<221 kW engine) and allowed to fish within 12 nm of the coast. Both pulse trawl (~ 1.1 tonne in air) and tickler trawl (~ 2.4 tonne in air) gears measured 4 m in width and had one trawl shoe at either side of its beam. The fishing vessels made a total of six trawl passes over each experimental plot at a speed of ~ 5 knots (pulse) and ~ 6 knots (tickler). Experimental fishing with the

pulse trawl was performed on 12 June 2018 between 4.30 and 11.30 AM UTC while the tickler trawl was used on 15 of June 2018 between 4.00 and 6.30 AM UTC (Table 1). Sonar, sediment profile imagery and benthic sample information was collected before/after trawling and in the reference areas (Fig. 1, Table 1). Water column information was collected during trawling activity. All experimental equipment was deployed from the research vessel (RV) *Simon Stevin*.

### 2.3 Water column measurements

To measure turbidity during trawling events, the RV deployed an SBE 25plus Sealogger (CTD) with two optical backscatter point sensors (OBS) while suspended continuously at 5 m depth during experimental trawling activity. For each treatment plot measured (A, B, C, D, E, G, and H), the RV strategically positioned itself against the current and the ongoing trawling ~ 80 m adjacent to the corresponding plots (due to logistic constraints plots F and I were not measured). Water was collected using Niskin bottles mounted on the CTD during Pulse-ON (plots B and C), Pulse-OFF (plots D and E), and Tickler (plots G and H)

treatments. Information for suspended particulate matter (SPM), particulate organic carbon (POC) and total nitrogen (TN) was obtained by running ~ 10 liters of collected water through precombusted GF/F filters (0.7 µm). Filters were dried at 60°C before being decarbonated with 0.1 N HCl (Nieuwenhuize et al., 1994) and analyzed with an Interscience Flash 2000 elemental analyzer.

### 2.4 Bathymetry and backscatter

Bathymetry and backscatter data were collected using a Kongsberg EM2040 single head multi-beam echo sounder (MBES) mounted on the RV (Depestele et al., 2018). While bathymetry data can tell us how sediment was deepened by trawling, backscatter (reflection of soundwaves) can give insight whether trawling affects sediment type or seabed roughness (Depestele et al., 2018). All data were collected with a MBES frequency of 300 kHz, a speed of 8 knots and at orientation parallel to the longest side of the experimental plots to avoid backscatter bias from the survey line directions (Lurton et al., 2018).

Bathymetrical and backscatter data were collected before (T0) and after (T1) trawling (Table 1). The RV followed the FV at an approximate distance of 300 m during one single haul in each of the experimental plots to register the bathymetrical data. After six passes in each experimental plot, the RV collected MBES backscatter data within 12 hours.

        A high resolution (0.5 x 0.5 m) digital elevation model of the seabed was created from the MBES data using Qimera (QPS). Trawl tracks were located from the GPS-derived position of the vessel path and visual detection using backscatter and

bathymetrical info. Transects of 12 m wide were positioned every 20 m along the trawl track in GIS software (N = 74 for





Pulse-ON, N = 100 for Pulse-OFF and N = 94 for Tickler; Fig. S2 in the Supplement). For the transects with visible trawl marks, the mean water depth was calculated for five locations inside and outside the trawl tracks. These measurements were based on the locations of the trawl track that were estimated from the MBES measurements at T1. The water depth was estimated at these same locations at T0. The difference between mean water depths inside and outside the track was used to

assess if the trawl caused bathymetrical changes in the track.

A backscatter mosaic (1 x 1 m) was created in FMGeocoder Toolbox (QPS FMGT). Backscatter data were sampled by (1) using transects for comparison of T0 and T1 and (2) by using all backscatter data in each experimental plot. Values outside 1.5 times the interquartile range were removed as outliers (1.5% of all values). This resulted in a total number of backscatter values > 120,000 for each treatment and time interval (T0 and T2).

**2.5 Sediment profile imagery**

Sediment Profile Imagery (SPI) was used to characterize surface and subsurface sediment features using a Nikon D7100 Digital Camera (Ocean Imaging Systems, North Falmouth, MA, USA). The SPI is a device (lowered onto the sediment from the RV) that captures images of the sediment-water interface in a vertical cross-section and can provide information on benthic fauna, physical disturbance, and chemical features (e.g. redox depth) within the upper sediment (Rosenburg et al., 2003). Three

replicate images were taken at each sampling station (3 stations per treatment plot; Fig. 1). One hundred eighty SPI photos were captured between T0-T1 treatment plots and reference stations. Images were taken in undisturbed sediments (T0) for the Pulse-ON treatment plots on 7 June 2018 (Table 1). On 8 June 2018, images were taken at T0 Reference areas and T0 Pulse-OFF plots. Images for two of the three Pulse-OFF T1 plots (D and F) were captured on 12 June 2018 1 – 2 h after trawling, though technical difficulties prevented further use of the SPI until the following day. The remaining T1 images for Pulse-ON

and Pulse-OFF treatments captured on 13 June 2018 (19 – 21 h after trawling) followed by visits to the Tickler (T0) plots. Tickler T1 images were captured on 15 June 2018, 1.5 – 3 h after beam trawling. Information for prism penetration depth (depth at which the camera penetrates the sediment) and visual observations of fauna, sediment characteristics (appearance of fine/coarse sediment, sand ripples etc.) and oxidation, expressed as % anoxic sediment (assessed by sediment color [black = anoxic]) were obtained using SpiArcBase software (Romero-Ramires et al., 2013). Individual SPI images can be found in Sect.

S4 of the Supplement.

**2.6 Benthic samples**

Samples for sediment, macrofauna, porewater nutrients, and biogeochemical incubations were collected using cylindrical NIOZ box corers (50 cm height, 32 cm diameter; Netherlands Institute for Sea Research, Texel). For timesteps T0 and T1, three replicate box cores were taken inside individual treatment plots (western, center, and eastern stations) and at reference

stations (69 box cores in total; Fig. 1). During experimental fishing activity, box cores in trawled sediments were collected shortly after 6 trawling consecutive passes were made in each treatment plot (within 0.5 h after fishing). On 11 June 2018, box



cores were collected for T0 Pulse-ON and Pulse-OFF treatments and at the reference stations (Table 1). T1 samples for Pulse-ON and Pulse-OFF treatments were collected on 12 June 2018. Box cores for T0 Tickler treatment plots were collected on 13 June 2018. Tickler T1 and Reference T1 samples were collected on 15 June 2018.

### 2.6.1 Sediment parameters


Box cores were subsampled for sediment at the western and eastern stations of each plot and for replicates 1 and 3 for reference stations. Sediment samples were subdivided at 0–0.5, 0.5–1, 1–1.5, 1.5–2, 2–2.5, 2.5–3, 3–4, 4–5, 5–6, 6–8, 8–10 cm slices taken from 3.5 cm inner diameter subcores. All samples were stored in a -20 °C freezer before freeze-drying and sieving (1 mm) for particle analysis. Sediment grain sizes were obtained via laser diffraction using a Malvern Mastersizer 2000 (McCave

et al., 1986). Chlorophyll *a* (chl *a*) pigments were extracted using acetone and analyzed with UV spectrophotometry (Ritchie, 2006).

To estimate bioturbation (faunal-mediated particle re-working) and/or physical sediment mixing after trawl disturbance $D_B$ values were calculated by fitting a biological mixing model to chl *a* profiles (Soetaert et al, 1996; Morys et al., 2016, Morys et al., 2017). $D_B$ (cm$^2$ d$^{-1}$) is often referred to as the "bioturbation coefficient", however, as we used this metric

to also estimate mixing from trawl disturbance, we refer to these values hereafter as the "sediment mixing coefficient". Values for $D_B$ were derived from one of the six bio-mixing models, as detailed in Soetaert et al, (1996). Model 1 describes sedimentation without mixing while model 2 assumes only "local" diffusive mixing. Models 3, 4a, 4b and 5 all pertain to "nonlocal" sediment mixing which occurs when organisms transport surface particles to deeper layers of the sediment. Model 3 includes an injection flux of material from the surface to a certain depth and model 4a adds the thickness of the injected

layer. Model 4b specifies the ingestion rates of material injected in depth while model 5 (the most complex) includes ingestion and thickness of the deposited layer. Selection of optimal model fits were made based on their Akaike's information criterion (AIC) scores and visual assessments.

### 2.6.2 Porewater nutrients

Two box cores were subsampled for porewater at western and eastern stations for each treatment plot and reference stations

(replicates 1 and 3) with 10 cm diameter sub cores with vertical sampling ports. Porewater was extracted at 0, 1, 2, 3, 5, 7 and 10 cm depth using rhizon samplers (Rhizophere Research Products; Seeberg-Elverfeldt et al., 2005; Dickens et al., 2007; Shobolt 2010). Porewater samples were added to 10 mL polystyrene vials and stored in -20 °C prior to analysis. To acquire nutrients, samples were thawed and analyzed using a SEAL QuAAtro segmented flow analyzer (Jodo et al., 1992) to determine concentrations of ammonium ($NH_4^+$), nitrite ($NO_2^-$), nitrate ($NO_3^-$), phosphate ($PO_4^{3-}$), and silicate ($Si(OH)_4$).

### 2.6.3 Benthic biogeochemical fluxes


Incubation cores (15 cm inner diameter by 30 cm height) were collected from all box core samples (3x per treatment plot and reference station) with overlying water. These were subsequently placed in a water bath filled with surface seawater and left to incubate to obtain information on nutrient fluxes exchanging to and from the sediment and water column in dark conditions. Magnetic stirrers, fixed to the lid of incubation cores, ensured homogenized water column conditions. Incubation cores were

sealed from air contact for 4 hours to measure rates of sediment community oxygen consumption (SCOC). During this period, oxygen concentrations were monitored every 30 s using optode sensors placed in the overlying water of the incubation cores (FireStingO$_2$, Pyroscience). At the end of the O$_2$ incubations, cores were re-opened and aerated to remain at above 90% O$_2$ saturation for the ongoing nutrient incubations. Ten mL water samples were collected at the overlying water of incubation cores at 0, 4, 8, 16 hour time intervals to obtain flux information on NH$_4^+$, NO$_2^-$, NO$_3^-$, PO$_4^{3-}$, and Si(OH)$_4$. Water samples for

nutrient fluxes were stored and analyzed using the same methods described for porewater nutrients. Flux estimates were obtained by fitting linear regressions on nutrient concentrations changes over time and multiplying the regression coefficient with the height of the overlying water (Tiano et al., 2019).

### 2.6.4 Mass budget modelling

Rates for total mineralization of organic matter, denitrification, nitrification and the percentages of nitrogen and phosphorous

removal were estimated using the measured sediment-water exchange fluxes of O$_2$, NH$_4^+$, NO$_2^-$, NO$_3^-$, PO$_4^{3-}$ calculated from the core incubations. These flux values were used to create an integrated mass budget of the solutes within the sediment column (Soetaert et al., 2001; Braeckman et al., 2010). More detailed methodology can be found in Sect. S5 in the Supplement.

### 2.6.5 Macrofauna collection

After the flux measurements, incubation cores were rinsed through a 1 mm sieve to collect the benthic macrofauna inside them.

Faunal samples were preserved in 8% formalin seawater on the RV. Sorting and identification of preserved macrofauna samples took place at the Flanders Institute for Agricultural and Fisheries Research (ILVO) to obtain measurements for taxon densities and biomass (blotted wet weights) following the accredited procedure: BELAC ISO17025 norm (ILVO-DIER-ANIMALAB; Certificate N°:BELAC T-315).

## 2.7 Statistical analysis

### 2.7.1 Water column

Differences between water column conditions (continuous CTD measurements and SPM filters) during Pulse-ON, Pulse-OFF and Tickler treatments were assessed with one way ANOVA's and Tukey HSD pairwise tests. Assumptions of normality and



homogeneity of variances were verified using Shapiro-Wilk and Levene's tests. If parametric assumptions were violated, data underwent a natural log-transformation prior to analysis.

### 2.7.2 MBES information

Bathymetrical differences between water depths inside and outside the track locations $\Delta_{I,T,G}$ were calculated for each transect $i$ at each time interval $T$ (T0 = before and T1 = after trawling) for each trawl gear type $G$ (1). A paired Wilcoxon signed-rank test was used to test if Delta-values were significantly different between T1 and T0 for each gear type. Significant positive Delta-values indicated that the water depth inside the trawl track was higher than outside the trawl track.

$$\Delta_{i,T,G} = (water\ depth_{inside} - water\ depth_{outside\ track\ location})_{i,T,G} \tag{1}$$

The indention $I$ of each transect $i$ was calculated by subtracting the delta value at T0 from the Delta-value at T1 (2).

$$I_{i,G} = (\Delta_{T1} - \Delta_{T0})_{i,G} \tag{2}$$

A resulting positive value indicated a deepening of the trawl track. The indention of the trawl was compared between trawl types using a Kruskal-Wallis Rank sum Test.

Statistical differences in backscatter values were compared for the combination of time and trawl types using experimental plots as replicates in a Kruskal-Wallis Rank sum Test. The experimental plots were also compared using the Kruskal-Wallis Rank sum Test. A Dunn test with Benjamini-Hochberg method was applied for pairwise comparisons between sites.

### 2.7.3 SPI and benthic sample analysis

Linear mixed effects models (LMM) were used to investigate SPI penetration depth, sediment, porewater nutrient concentrations, biogeochemical flux, mass budget model results and ecological characteristics (individual densities, biomass, species richness) using the *lmer*-function in the R package: "lme4" (Bates et al., 2015). For each treatment, a "model-a" was created to include "timestep" (T0 or T1) as a fixed variable, "temperature" as a co-variate, and "station" as a random variable to minimize spatial autocorrelation between sample locations. A "model-b" considered only the random variable (station). Model-a and model-b were tested against each other using a partial F test.

### 2.7.4 Impact of *Lanice conchilega*

Robust regression analysis, using the R package: "robustbase" (Maechler et al., 2020), was conducted to investigate the effect of *L. conchilega* densities and biomass on physical, biogeochemical and faunal parameters in trawled and untrawled samples. This alternative to least squares techniques is an iterative analysis which down-weighs the influence of outliers on regression





coefficients and is less affected by violations of linear regression assumptions (Koller and Stahel 2011). To reduce possible bias from the spatial differences in *L. conchilega* densities/biomass between treatment plots, all pre-trawled and post-trawled data were pooled together for this analysis.

All data visualization and statistical analyses were carried out using R (R Core Team 2021).

## 3 Results

### 3.1 Water column conditions during trawling

Turbidity measured by the OBS sensors detected significantly higher amounts of water column SPM concentrations (mg L$^{-1}$) during tickler trawling (mean ± standard deviation = 62.5±17.1, $p < 0.001$) compared to Pulse-ON (19.6±5.3) and Pulse-OFF treatments (13.3±5.8; Fig. 2). SPM filters collected higher average estimates of water column turbidity (Tickler = 100.6±18.7; Pulse-ON = 21.6±1.7, Pulse-OFF = 21.8±1.6) with significantly higher turbidity during the Tickler treatment compared to both

pulse fishing treatments (p < 0.01). Higher SPM on the filters coincided with lower POC (%) (Tickler = 2.4±0.2; Pulse-ON = 3.6±0.1, Pulse-OFF = 4.3±0.6) and TN (%) percentages (Tickler = 0.4±0.01; Pulse-ON = 0.7±0.2, Pulse-OFF = 0.6±0.05) with the Pulse-OFF treatment showing a significantly higher water column POC percentage compared to the tickler trawling ($p <$ 0.05). Data from a "Meetnet Vlaamse Banken" buoy (located in ~ 7 km southwest of the experimental location) showed current speeds ranging from 0.08 to 1.21 m s$^{-1}$ during pulse trawling and 0.23 to 0.85 m s$^{-1}$ during tickler trawling. On the day before

both pulse trawl treatments, water currents reached a maximum speed of 1.23 m s$^{-1}$ compared to 1.5 m s$^{-1}$ the day before the tickler treatment (Fig. S3 in the Supplement).

### 3.2 Trawl effects on bathymetry

Delta-values were significantly different before and after trawling for each trawl type (Pulse-ON: $p < 0.05$, Pulse-OFF: $p$ <0.001, Tickler: $p$ <0.001; Fig. S4 in the Supplement), though the effect sizes were relatively small (Cohen's d estimates of -

0.21, -0.30 and -0.29 for Pulse-ON, Pulse-OFF and Tickler respectively). The track locations were deepened, on average, by 1 cm after trawling (Pulse-ON: 0.99±3.92; Pulse-OFF:1.17±1.84; Tickler: 1.11±2.74). The indention of the trawl track was not significantly different between treatments.

### 3.3 Trawl effects on backscatter

The mean backscatter values of the track locations in all treatments exhibited reduced values after 1 trawl pass (ΔT0 – T1:

Pulse-ON = -0.20; Pulse-OFF = -0.23; Tickler = -0.30) though there was no statistical difference between the three trawl types (Table S1 in the Supplement). Comparison of mean backscatter values over entire plots (after 6 trawl passes) showed dB values ranging between -26.64 and -25.40 (Table S2 in the Supplement). The mean backscatter strength did not differ by more than



2 dB between the treatments at T0 and T1 and was not significantly different between any of the time-treatment interactions (Fig. 3). Backscatter values were not statistically different by experimental plots (Fig. 3). Uncorrected *p*-values showed
significant pairwise differences (p < 0.05) for plot A and plots: B, H, I, and also with plot F and plots: B, C, H and I (Fig. 3).

**3.4 Sediment profile image analysis**

*L. conchilega* tubeworms were identified in 72 % of images taken and were often associated with heterogeneous patterns of oxidized sediment (Fig. 4b). Mechanical disturbance was evident in certain T1 images showing "flattened" tubes and/or sediment (Fig. 4c). The percentage of anoxic sediment was significantly lower in T1 images from the tickler treatment
compared to T0 images ($p < 0.01$) but did not differ significantly from T0 – T1 in other trawl treatments (Table 2). Anoxic (dark) sediment was significantly higher in images from reference areas taken at the end of the experimental period (T1) compared to the beginning (T0; $p < 0.001$). The sediment penetration depth of the SPI prism did not differ significantly between T0 – T1 for any treatment (Table 2).

**3.5 Trawl effects on sediment parameters**

Very fine sand fractions (62.5 – 125 µm) and chl *a* content in the sediment column (0 – 10 cm) were significantly lower in Tickler T1 compared to T0 samples ($p < 0.01$; $p < 0.001$; Table 2). For the Pulse-ON treatment, T1 silt fractions and chl *a* content were significantly lower than before trawling ($p < 0.01$; Table 2). Surface sediment (0 – 1 cm) showed significant increases in median grain size (µm) between T0 – T1 for Tickler (T0 = 186±75, T1: 227±71; $p > 0.01$) and Pulse-ON (T0 = 209±50, T1: 243±62; $p < 0.05$) treatments. Larger grain sizes in surface sediment coincided with significantly smaller silt (%)
fractions (Tickler: T0 = 16.8±14.7, T1= 6.1±8.1, p < 0.001; Pulse-ON: T0 = 9.4±7.2, T1 = 5.1±8.8, p < 0.05) and chl *a* (µg g⁻¹) content (Tickler: T0 = 20.0±16.2, T1 = 7.4±10.8, $p < 0.001$; Pulse-ON: T0 = 13.7±9.2, T1 = 8.6±12.5, $p < 0.05$) in the same sediments. Robust regression of chl *a* content versus silt percentages in the sediment column explained 92 % of the variation. No significant differences in sediment parameters between T0 – T1 were found in the Pulse-OFF treatment. Particle size distributions of surface sediments (0 – 1 cm depth) for the Tickler treatment showed a decrease in fine sediment fractions along
with an increase in the percentage of coarser particles (Fig. 5). The Pulse-ON treatment was characterized by a shift towards larger grain size fractions after trawling. In contrast, no conspicuous shifts in particle size distributions were detected in Pulse-OFF or Reference areas (Fig. 5).

Nonlocal mixing was detected in 89% of chl *a* profiles analyzed. Mean sediment mixing rates ($D_B$) increased between T0 – T1 for all trawled treatments, however, no changes were statistically significant (Table 2). Reference areas displayed
higher mean mixing rates compared to sediments in all trawl treatment plots.





### 3.6 Trawl effects on biogeochemical fluxes and nutrient profiles

Average SCOC rates (mmol $O_2$ m$^{-2}$ d$^{-1}$) were significantly reduced between T0 (172±109) and T1 (74±40) for the Tickler treatment ($p < 0.001$; Fig. 6a; Table 3). No other consistent or significant patterns were found for nutrient fluxes between T0 – T1 fishing treatments. Mean ammonium fluxes (mmol $NH_4^+$ m$^{-2}$ d$^{-1}$) ranged from 0.5±0.3 at Pulse-OFF T0 to 1.6±1.6 at
Tickler T1. Nitrate fluxes (mmol $NO_3^-$ m$^{-2}$ d$^{-1}$) were highest at Pulse-ON T1 (0.02±0.02) and were directed into the sediment for reference samples (-0.02±0.03). Fluxes of phosphate (mmol $PO_4^{3-}$ m$^{-2}$ d$^{-1}$) ranged from 0.01±0.03 (Pulse-ON T0) to 0.05±0.07 (Pulse-OFF T0). Silicate effluxes (mmol $Si(OH)_4$ m$^{-2}$ d$^{-1}$) were lowest at Pulse-ON T0 (0.7±0.6) and highest at Tickler T1 (1.7±0.7; Table S3 in the Supplement).

Average porewater nitrate concentrations within the sediment column (0 – 10 cm) were significantly higher after
trawling in Tickler and Pulse-OFF treatments compared to pre-trawled sediments (Tickler, $p < 0.01$; Pulse-OFF, $p < 0.01$; Table 3). Ammonium concentrations in the sediment significantly increased from T0 to T1 ($p < 0.01$) after tickler trawling (Table 3). Mean phosphate concentrations were elevated after trawling for all experimental treatments but this was significant only in Pulse-ON ($p < 0.01$) and Pulse-OFF ($p < 0.001$; Table 3).

### 3.7 Mass budget calculations

The Tickler treatment exhibited significantly lower values for total mineralization ($p < 0.001$), denitrification ($p < 0.05$), and nitrification ($p < 0.05$) in T1 compared to T0 samples (Table 3). The percentage of nitrogen removed after mineralization ranged between 80 and 95 % while sedimentary phosphorus removal varied between 91 and 99 %, however, no changes between T0 – T1 were statistically significant for individual fishing treatments. Total organic matter mineralization was significantly higher in T1 reference samples compared to T0 ($p < 0.05$; Table 3).

### 3.8 Trawl effects on macrofauna density and macrobenthos composition

Sixty-nine taxonomical groups were identified in this study with *L. conchilega* dominating benthic macrobenthos densities (54%). The bivalve, *Kurtiella bidentata*, and polychaetes, *Magelona spp.*, and *Eumida sanguinea* made up 13%, 4% and 3% of total macrofauna density respectively. Though they only contributed 1% of total macrobenthos density, the razor clam, *Ensis leei*, represented 45% of total biomass collected followed by *L. conchilega* (22%), *Spisula subtruncata* (bivalve, 14%),
and *Echinocardium cordatum* (heart urchins, 4%).

Total macrofaunal densities (individuals m$^{-2}$) decreased from T0 (Tickler: 11,782±7,360; Pulse-ON: 9,783±14,072; Pulse-OFF: 6,241±5,120) to T1 (Tickler: 4,949±3,428; Pulse-ON: 3,336±3,529; Pulse-OFF: 5,483±3,623) but were only statistically significant after the Tickler treatment ($p < 0.05$; Fig. 6b). Reference samples varied greatly but held the highest average densities from all the treatments (T0 = 19,174± 17,493; T1 = 22,993± 17,945; Fig. 6b).





*L. conchilega* maintained the highest average densities (individuals m$^{-2}$) in the reference stations (T0 = 10,703±9,943; T1 = 14,707±11,336). *L. conchilega* densities declined between T0 and T1 for Pulse-ON (T0=5,801±9,352; T1 =1,587±3,022) the Pulse-OFF (T0 = 3,081±3,711; T1 = 2,251±3,547) and Tickler treatments (T0 = 5,880±3,838; T1 = 2,172±1,913), but was only significant for the Pulse-ON treatment ($p < 0.05$). Amongst other abundant taxa, *Abra alba* (bivalve) density declined significantly after tickler trawling (T0 = 418±632; T1 = 231±420, $p < 0.05$) and *E. sanguinea* was significantly lower after the

Pulse-ON treatment (T0 = 555±1,315, T1 = 93±195, $p < 0.001$).

### 3.9 Effects of *Lanice conchilega* in trawled and untrawled sediments

*L. conchilega* was found in 68 out of 69 (98.5 %) benthic sediment samples collected in this study. In pre-trawled sediments, *L. conchilega* density explained 85 % of variation in the abundances of the other taxa ($p < 0.001$) and while biomass explained 38 % of variation in species richness ($p < 0.001$; Fig. 7, Table 4). Tubeworm densities in pre-trawled samples were linked with

smaller sediment grain sizes, accounting for 33 % of variability in the percentage of very fine sand ($p < 0.05$) while biomass explained 18 % of variability in silt content ($p > 0.01$) (Table 4). Densities of *L. conchilega* significantly predicted 87 % of SCOC in undisturbed sediments ($p < 0.001$). Biomass of *L. conchilega* (compared to density) was a slightly better predictor of SCOC and explained 93 % of variation in O$_2$ fluxes ($p < 0.001$) as well as 21 % of nitrate effluxes ($p < 0.01$) in untrawled sediments (Fig. 7; Table 4). No significant relationships were found after trawling (Fig. 7, Table 4). Due to lack of sufficient

data, regressions between T0 – T1 within individual fishing treatments (Pulse-ON, Pulse-OFF, Tickler) did not yield consistent results.

### 4 Discussion

The data gathered in this study exhibited a large amount of spatial variability reflecting the dynamic nature of the Vlakte van de Raan ecosystem (Degraer and Hostens, 2016; Fig. 3). Despite the challenge of detecting disturbance in such a variable

system, we were able to provide a comprehensive assessment of acute bottom trawling with our 3 x 3 treatment plots and BACI design. The dominance of *L. conchilega* in this habitat provided an additional opportunity to explore the effects of tubeworm abundance on environmental parameters in disturbed and undisturbed conditions. Here, we discuss water column observations, trawl-induced changes to physical, biogeochemical, and ecological characteristics followed by implications for fisheries management and general conclusions.

This study follows up on similar research conducted in the Frisian Front region of the North Sea (Depestele et al., 2018, Tiano et al., 2019; Tiano et al., 2020) with each study (including the current one) collecting information in the same season (early June) of its respective year. The offshore Frisian Front ecosystem (~ 60 km from the Dutch Coast), however, experiences much lower levels of tidal disturbance, exhibits smaller grain sizes, and holds completely different macrofaunal assemblages and biogeochemical characteristics compared to the Vlakte van de Raan (van Raaphorst et al., 1992; Dewicke et



al., 2002; Degraer and Hostens, 2016; Depestele et al., 2018). By comparing the findings from these two contrasting areas, we can better understand how different ecosystems are affected by bottom disturbance. To our knowledge, this is the first coastal habitat study that simultaneously investigates acute beam trawl impacts on biological (Rabaut et al., 2008), physical (Depestele et al., 2016) and biogeochemical dynamics.

### 4.1 SPM during experimental trawling

The water column was 73-78% more turbid during tickler trawling compared to both pulse trawl treatments. As tickler trawls tow at faster speeds compared to pulse trawls (Poos et al., 2020), they are predicted to mobilize more sediment (Rijnsdorp et al., 2021). However, due to the higher current speeds evident in the day prior to tickler trawling, we cannot be certain that this increased turbidity was the result from enhanced resuspension caused from bottom trawling alone. It is possible that trawling coupled with current induced resuspension enhanced SPM measurements during the Tickler treatment. This would corroborate

with the findings in Depestele et al., (2016) where no significant differences in sediment resuspension of pulse and tickler trawling was found in a similar coastal area. In the Frisian Front, where Tiano et al., (2019) reported conspicuous SPM peaks attributed to individual trawl passes, water current speeds were 62 – 88 % lower during fishing compared to the Vlakte van de Raan study. The more ambiguous link between trawling and SPM in our study (no obvious turbidity peak during trawl passes) suggests that the water column effects of trawling are more difficult to detect in areas with higher natural current speeds.

### 4.2 Changes to the physical environment

A single trawl pass for all three treatments yielded a bathymetrical indentation of ~ 1 cm in the sediment, similar to the 0.9 cm found after tickler trawling in coarse sediment off the Dutch coast (Depestele et al., 2016). Our study, however, did not replicate previous findings, which found greater bathymetrical effects for tickler beam trawling in finer North Sea sediments (Depestele et al., 2018). This is likely due to differences in grain size which are smaller in the Frisian Front (aforementioned study),

compared to the Vlakte van de Raan (current study). Backscatter data in our study produced similar results (after 1x and 6x trawl passes) between treatments, though spatial differences were detected in the northern and southernmost plots (Fig. 3). Trawl marks in sandy seabed's typically reduce the backscatter value, causing more negative dB values (Roche et al., 2018). The backscatter values of the track locations in our study were consistently reduced, but the magnitude of the changes was, on average, too low to be considered as a substantial alteration in backscatter value due to trawling (Roche et al., 2018).

Backscatter values can approximate sediment characteristics, but do not purely reflect sediment grain size and can be distorted by "scatterers" such as shell debris and biogenic reefs, which may interfere with the acoustic signals (Degraer et al., 2008; Feldens et al., 2018).

Tubeworm reefs in untrawled sediments were positively associated with smaller grain size fractions (Fig. 4; Table 4). This pattern is commonly found with *L. conchilega* as high tube densities will slow down bottom current velocities leading to

the settling of smaller particles (Van Hoey et al., 2008; De Smet et al., 2015; Foshtomi et al., 2018). Trawl induced coarsening



of sediments found in *L. conchilega* dominated habitats may occur through both the resuspension and winnowing of finer grained particles from direct trawling impacts (Palanques et al., 2014; Mengual et al., 2016; Depestele et al., 2018; Tiano et al., 2019) and the alteration of sediment characteristics caused by the rapid extraction of tube structures from the seabed.

The shift in particle sizes, changes in observed anoxic sediment and the average increase in sediment mixing rates after trawling, suggests that the largest sediment alterations were attributed to the Tickler treatment followed by the Pulse-ON treatment with no significant effects detected after Pulse-OFF trawling (Fig. 5; Table 2). As electricity does not affect physical sediment characteristics (Tiano et al., 2021), differences between Pulse-ON and Pulse-OFF treatments may have been due the variable mechanical impacts of pulse trawls (Depestele et al., 2018) or the environmental variability of the site (Degraer and Hostens, 2016). Compared to trawl treatment plots, sediment mixing rates were higher, on average, in reference sites (Table 2) likely due to higher abundances of macrofauna (more bioturbation) associated with those areas (Fig. 6b). For all trawled areas, mean sediment mixing rates increased from T0 to T1 despite reduced macrofaunal densities (Table 2, Fig. 6b). This suggests that physical trawling effects on the sediment may function similarly to high levels of bioturbation (Duplisea et al., 2001) and may thus result in the burial of some organic matter and nutrients (Mayer et al., 1991).

**4.3 Changes to biogeochemical characteristics**

We detected significant declines in chl *a* from surface sediment (0 – 1 cm) of 63% and 37% after tickler and pulse trawling (Pulse-ON) respectively. These percent changes are lower relative to the reductions measured after tickler (83 %) and pulse (43 %) trawling in the siltier Frisian Front sediments (Tiano et al., 2019) even with the three to six times higher sediment chl *a* content in the more coastal Vlakte van de Raan system. Sandy dynamic habitats, like the Vlakte van de Raan, experience higher bed shear stress and are generally less vulnerable to trawling impacts compared to muddier more static environments like the Frisian Front (Kaiser et al., 2006; Queirós et al., 2006; Allen and Clarke, 2007; Sciberras et al., 2016; Rijnsdorp et al., 2020a). Moreover, this was demonstrated with the reduced bathymetrical alteration caused from trawl passes found in the current study compared to pulse and tickler trawling in the Frisian Front (Depestele et al., 2018).

Despite the lower mechanical impact (~ 1 cm Vlakte van de Raan vs. ~ 4 cm Frisian Front; Depestele et al., 2019), our study produced a stronger statistically significant reduction in SCOC (Tickler: 57 %) compared to Tiano et al., (2019; Tickler: 41 %, Pulse: 33 %). Significant reductions in total mineralization, denitrification, and nitrification after the Tickler treatment suggests that trawling reduced benthic nutrient removal and recycling functions in this study. Furthermore, SCOC and mass budget estimates suggest an average increase in benthic biogeochemical processes in the reference areas during the experimental period (Fig. 6a, Table 3). If these reference results provide an accurate representation of untrawled environmental conditions during the experiments, this may imply that the trawling effect on biogeochemical parameters measured in this study (decreased biogeochemical functioning) is conservative given the increased metabolism in the reference sites.

The large changes in SCOC found in this study may be explained by the decrease in fauna such as *L. conchilega*, which can be highly influential to SCOC and other biogeochemical fluxes (Braeckman et al., 2010). In addition to respiration,





when *L. conchilega* (and other infauna) irrigate their burrows, they can drastically expand the surface area available for sediment-oxygen exchange (De Smet et al., 2016b; Kristensen and Kostka, 2013), with major impacts to biogeochemical

dynamics (Olsgard et al., 2008; Braeckman et al., 2010). These results also suggest that areas with *L. conchilega* may be vulnerable to relatively shallow (~ 1 cm) seabed disturbances as their tubes extend above the sediment surface. Contrary to this, faunal-mediated biogeochemical functions in the Frisian Front are more relegated to organisms residing deeper within the sediment and are probably less affected by trawling despite having softer sediments and greater trawl penetration (Depestele et al., 2019; Tiano et al., 2019, 2020).

*L. conchilega* in untrawled sediments had a clear effect on $O_2$ and $NO_3$ fluxes, though these relationships disappeared after disturbance (Fig. 7, Table 4). The change in these relationships probably came from a combination of reduced average tubeworm densities (consistently lower after trawling but only significant after the Pulse-ON treatment) and direct trawl-induced habitat alterations. These results support the assertion by Hale et al., (2017), that trawling decouples faunal-mediated biogeochemical processes.

The Pulse-ON treatment, which led to the biggest decrease (72%) and lowest *L. conchilega* densities after T1, was also the only experimental treatment that did not produce a significant increase in porewater nitrate concentrations. Previous research suggests a density dependent enhancement by *L. conchilega* on denitrification/nitrification processes (Aller, 1988; Braeckman et al., 2010; Foshtomi et al., 2018). Trawling has also been shown to decrease denitrification by altering the sediment layers that optimize denitrification and reducing total mineralization rates (Ferguson et al., 2020; De Borger et al.,

2021). It is possible that both mechanical trawl disturbance and a loss (Pulse-ON) or retention (Pulse-OFF, Tickler) of functional species roles after trawling led to the enhanced sediment nitrate concentrations, though it is difficult to pinpoint the exact cause of this increase.

### 4.4 Benthic community observations

Our study found a significant decrease in *L. conchilega* densities with the Pulse-ON treatment and not in the other trawl

treatments, however, it is unlikely that electrical stimulation caused this reduction. Previous research has not found any direct mortality of fish or benthos associated with electrical stimulation used in the pulse trawl fishery (Polet et al., 2005; van Marlen et al., 2009; Soetaert et al., 2015b; Soetaert et al., 2016; ICES, 2018, 2019, 2020; Boute et al., 2020). Moreover, electric stimulation from flatfish pulse trawls cannot physically remove *L. conchilega* tubes from the sediment (Boute et al., 2020; ICES 2020; Tiano et al., 2021). Our results, however, point to higher physical disturbance occurring in the Pulse-ON areas as

being the likely culprit for reduced *L. conchilega* densities (Fig. 5, Table 2).

The Pulse-ON treatment also caused a significant decrease of *Eumida sanguinea*, a species closely associated to *L. conchilega* (Rabaut et al., 2008) though other taxa, such as *Abra alba*, only showed notable decreases in response to tickler trawling. Nonetheless, when accounting for total macrobenthos densities, the only significant effect came from the Tickler treatment (58% decrease) reflecting the more consistent effect associated with tickler beam trawl gears (Depestele et al., 2018;





Tiano et al., 2019). Pre-trawled habitats in this study linked *L. conchilega* to the densities of other taxa and species richness, though this relationship was no longer detected after experimental trawling (Fig. 7, Table 4). This suggests that, although tubeworm densities were impacted to some extent, trawling disproportionally affected the corresponding taxa. Rabaut et al., (2008) reported significant changes to an intertidal *L. conchilega* dominated community after beam trawling but not to the tubeworms densities themselves. Our results show a similar removal of species associated with *L. conchilega* (*E. sanguinea*

in particular), though, our findings of decreased tubeworm densities support evidence that *L. conchilega* shows a certain level of sensitivity to physical sediment disturbances (Callaway et al., 2010). *L. conchilega* has been found to successfully recolonize previously disturbed habitats within 1-2 years suggesting high resilience but not necessarily resistance to anthropogenic impacts (Callaway et al., 2010).

## 5 Conclusions

This study found various effects from conventional tickler rigged beam trawls and electric pulse trawl techniques. Though pulse trawls (and any other bottom fishing technique) can undoubtedly cause significant habitat alterations, these effects tend to be quite variable (Tiano et al., 2019 and this study). The more consistent effect of tickler trawls are, on average, higher than that of pulse trawls as shown in the sediment, geochemical, and species community level changes seen in this study. However, we also show that pulse trawls may be at least equally as impactful to densities of certain benthos. Bergman and Meesters,

(2020) and Depestele et al., (2019) found a reduced effect of pulse trawls compared to beam trawls, on benthic megafauna in the North Sea Oyster Grounds area and on the sediment properties in the Frisian Front respectively, while Tiano et al., (2020) found similar effects between the two gears on benthic infauna in the Frisian Front. It is likely that the lower average penetration depth of the lighter pulse trawl gears are less damaging to certain benthic communities compared to tickler trawls (Hiddink et al., 2017; Depestele et al., 2018), however, this reduced effect is dependent on species assemblages and sediment type.

Compared to any difference in direct impacts between pulse and beam trawls (due to different sediment penetration depths) the more important (relative) environmental benefit of pulse trawls likely stems from its higher catch efficiency which leads to reduced time needed to catch their quotas (less time fishing) when exploiting North Sea sole (*Solea solea*; Rijnsdorp et al., 2020b; ICES, 2020).

The high SCOC values found at our experimental location imply that this community is an important site for carbon

degradation (Stratmann et al., 2019). The trawl-induced declines in SCOC and total mineralization suggest that some disturbed locations in this community lost over half of their ability to mineralize organic matter. This highlights the propensity for bottom trawling to reduce carbon cycling in marine sediments and shift the mineralization processes to the overlying water (Tiano et al., 2019). With enhanced mineralization rates in the water column (due to higher oxic conditions compared to inside the sediment), trawling may lead to the removal of sedimentary carbon, increased nitrogen retention in the seabed (decreased

denitrification), and a reallocation of nutrients away from trawled sediments (Tiano et al., 2019; Ferguson et al., 2020; De Borger et al., 2021; Tiano et al., 2021).

Results in this study show that even highly dynamic coastal areas can be susceptible to significant trawl-induced changes depending on the type of benthic community present. Biogenic reef communities like the ones found in the Vlakte van de Raan, can be highly productive habitats with enhanced biogeochemical characteristics. With these features, the potential

for disturbance-induced changes becomes greater if keystone species such as *L. conchilega* are affected. When assessing the vulnerability of marine habitats, environmental managers and policy makers need to consider abundant ecosystem engineers as they perform important ecological and biogeochemical functions.

**Data availability**

Data and R code can be made available on request to the corresponding author.

**Author contribution**

JT, JD, GvH, and KS devised the study, and contributed to the manuscript. JT, PvR and KS collected and analyzed data for water column, nutrient flux, porewater nutrients, and sediment. JD, GvH and JF collected and analyzed data for sediment profile imagery, macrofauna, and sonar (bathymetry and backscatter).

**Competing interests**

GvH was a member of the editorial board for this special issue of Biogeosciences: "Towards an understanding and assessment of human impact on coastal marine environments", however, the peer-review process was guided by an independent editor, and no other authors have competing interests nor financial benefits to declare resulting from the publication of this manuscript.

**Acknowledgements**

*We would like to thank the crews of the TH6 for YE238 for their excellent work and coordination in making multiple trawled*
*areas. We extend a huge thank you to the crew of the Simon Stevin for helping us collect our data and putting up with our annoying requests to do more sampling and the students at ILVO who helped process our macrofauna samples. This work would not have been possible without the collaboration with the Flanders Marine Institute (VLIZ) who provided the RV Simon Stevin, along with the multiple data collection devices (multi-beam, CTD, SPI etc.). Finally, we send a special thank you to Peter van Breugel, Yvonne Maas, Jan Peene, Jurian Brasser, Yke Timmermans and any other members of the NIOZ analytical*
*lab for processing our sediment and biogeochemical samples. This research is partially funded by the European Maritime and Fisheries Fund (EMFF) and the Netherlands Ministry of Agriculture Nature and Food Quality (LNV).*

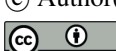



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



**Figures**

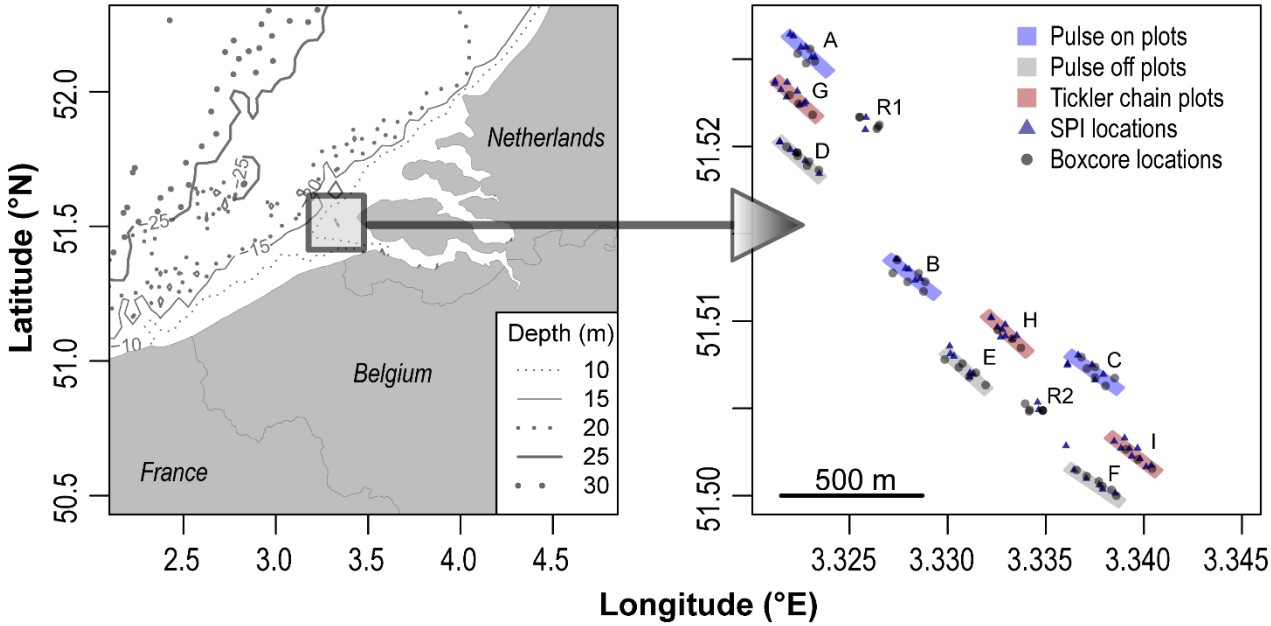

**Figure 1: Geographic location of the study site (left). Visual descriptions for experimental plots in Pulse-ON (A-C), Pulse-OFF (D-F), and Tickler (G-I) trawl treatments, reference stations (R1, R2), sediment profile imagery (SPI) locations, and box core sampling locations (right).**



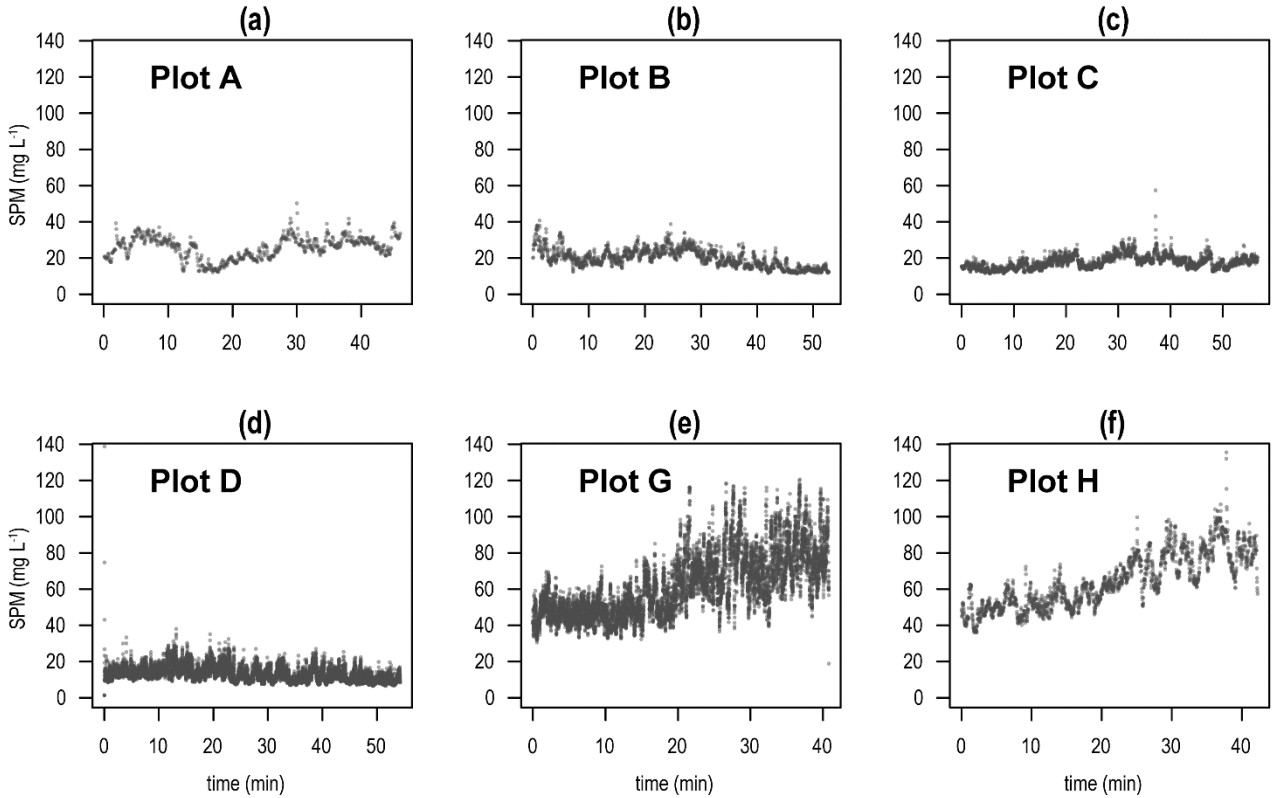


**Figure 2: Water column turbidity during the time of trawling for Pulse-ON plots (a-c), Pulse-OFF plot (d) and Tickler plots (e-f).**



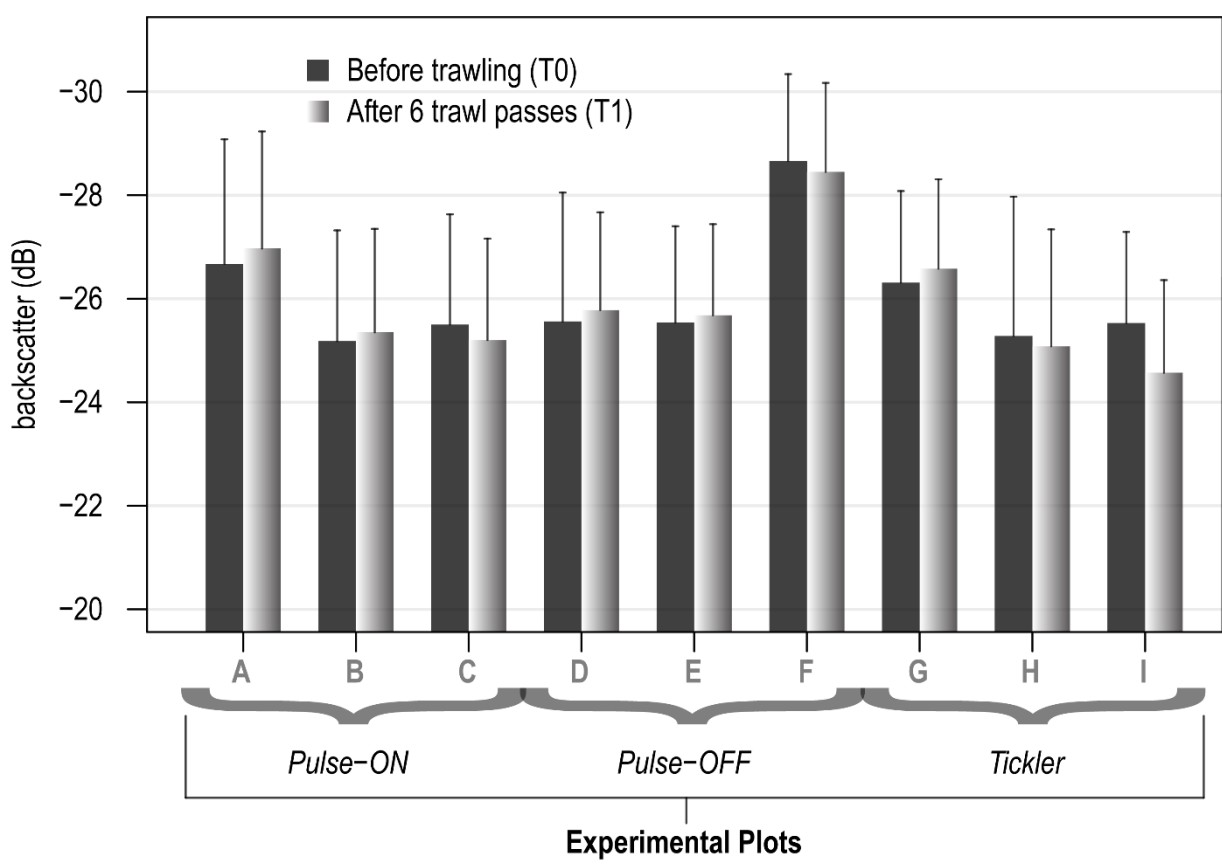

**Figure 3: Mean backscatter values (dB) per experimental plot (before and after trawling) with the standard deviations.**



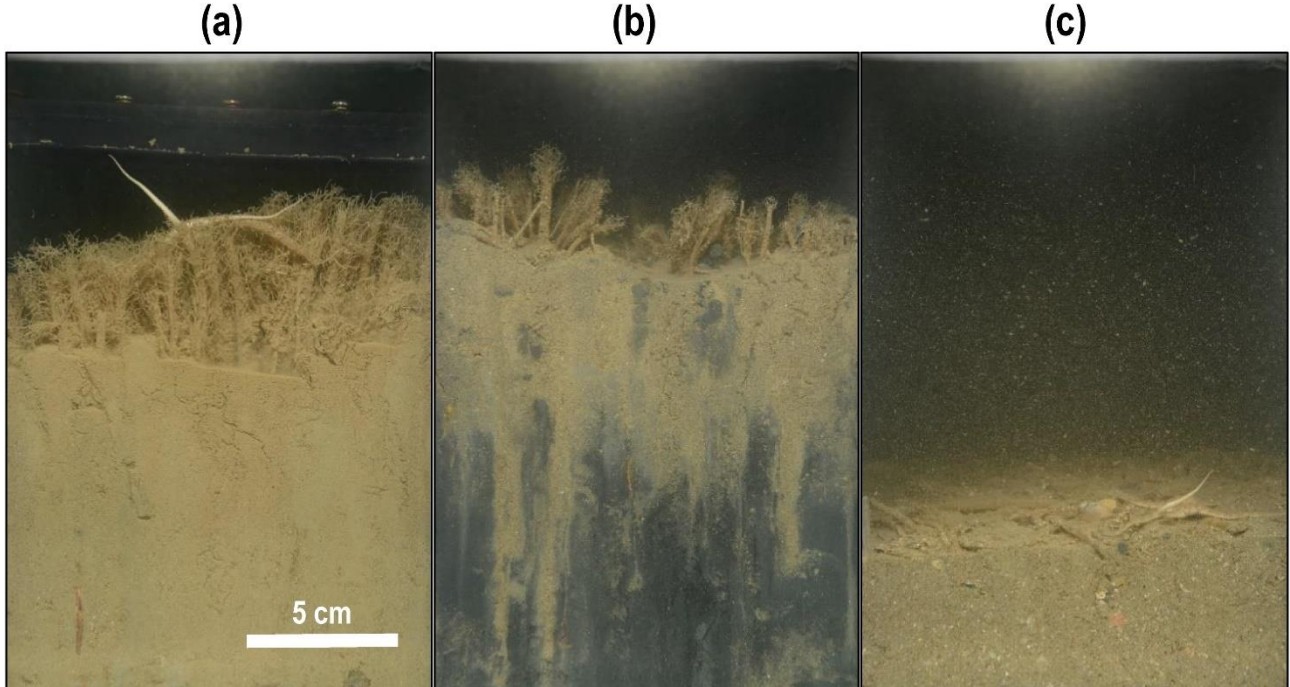

**Figure 4: Sediment profile imagery showing a dense *Lanice conchilega* tubeworm reef (a), bioirrigation-induced sediment oxidation (b) and evidence of trawl disturbance: flattened tubeworms and sediment (c).**



**Figure 5: Particle size distributions for the surface sediments (top 1 cm) in experimentally trawled samples. Solid lines**
**represent the distribution of sediment size fractions in Pulse-ON, Pulse-OFF, Tickler, and Reference areas before trawling (T0) and dashed lines represent distributions after trawl disturbances (T1).**



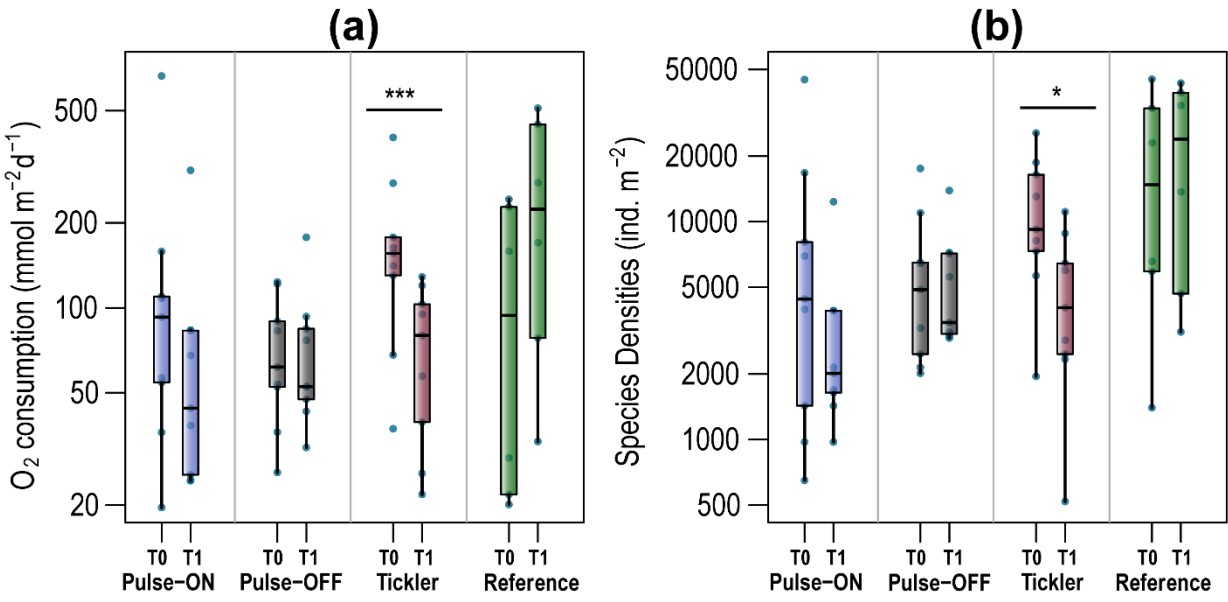

**Figure 6: Oxygen consumption (a) and macrobenthos densities (b) measured from treatment plots before (T0) and after (T1) fishing and in reference stations. Y-axes are on a log-scale.**



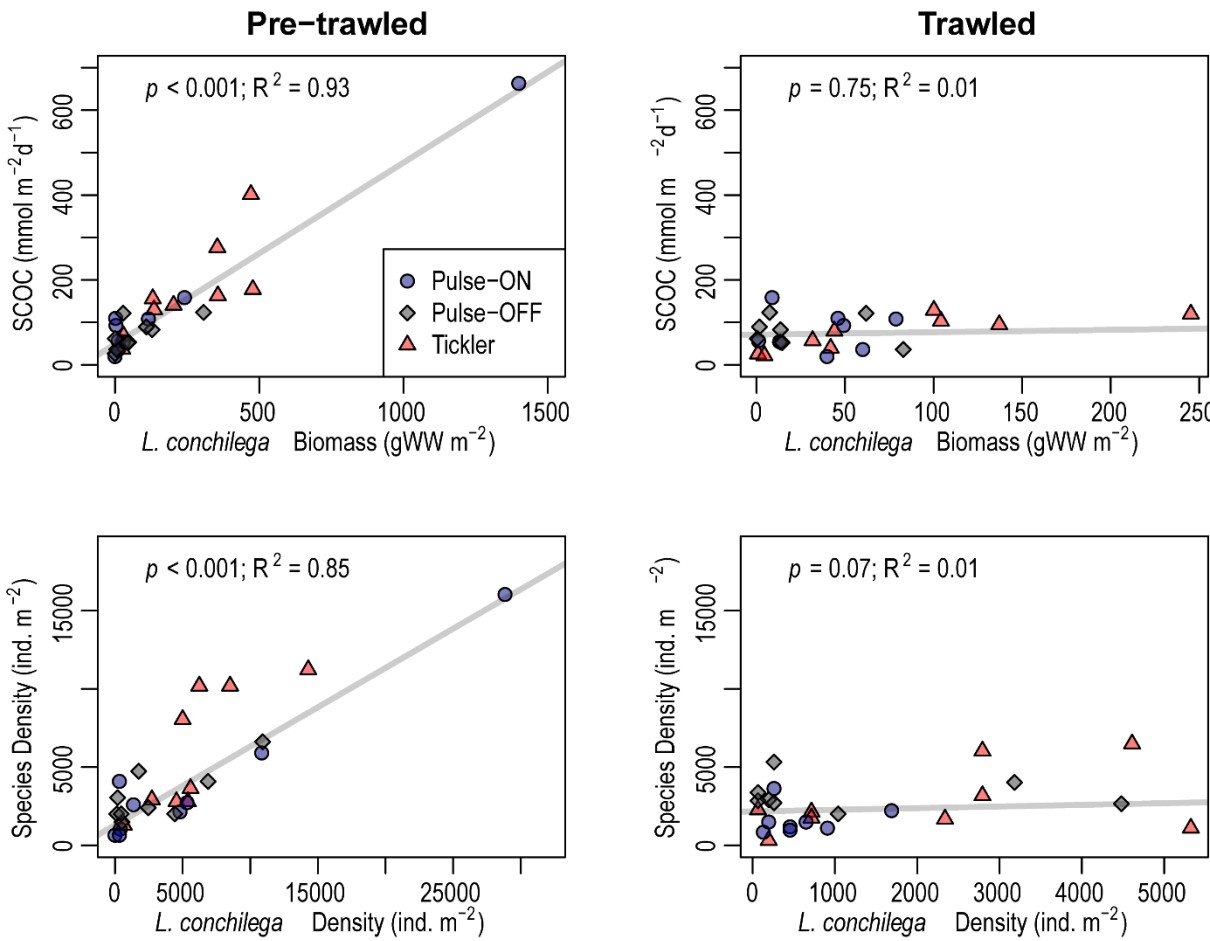


**Figure 7: Robust regression analysis of sediment community oxygen consumption (SCOC; mmol m$^{-2}$ d$^{-1}$) versus *Lanice conchilega* biomass (grams of wet weight m$^{-2}$) (top) and densities of other taxa versus *L. conchilega* densities (individuals m$^{-2}$) (bottom) in pre-trawled and trawled sediments. Note x-axis scales are different.**





**Tables**

**Table 1: Sampling information. Activities include experimental trawling, sediment profile imagery (SPI), sediment and macrofauna sampling (Box Core), bathymetry and backscatter (Sonar) and water column turbidity and organic matter measurements (Water Column). Benthic samples from box cores were also used to obtain porewater nutrients and biogeochemical flux data. Figure 1 displays the sampling locations specified in the "Areas" column.**

| Date | Activity | Treatment & Timestep | Areas |
|------|----------|---------------------|-------|
| *07 June 2018* | SPI | Pulse-On T0 | Plots: A, B, C |
| *08-Jun-18* | SPI | Pulse-OFF T0 | Plots: D, E, F |
| | | Reference T0 | R1, R2 |
| *11 June 2018* | Box Core | Pulse-On T0 | Plots: A, B, C |
| | | Pulse-OFF T0 | Plots: D, E, F |
| | | Pulse-On T0 | Plots: A, B, C |
| | | Pulse-OFF T0 | Plots: D, E, F |
| *12 June 2018* | Experimental Trawling | Pulse-On T1 | Plots: A, B, C |
| | | Pulse-OFF T1 | Plots: D, E, F |
| | Sonar | Pulse-On T1 | Plots: A, B, C |
| | | Pulse-OFF T1 | Plots: D, E, F |
| | Water Column | Pulse-On T1 | Plots: A, B, C |
| | | Pulse-OFF T1 | Plots: D, E, |
| | Box Core | Pulse-On T0 | Plots: A, B, C |
| | | Pulse-OFF T0 | Plots: D, E, F |
| | SPI | Pulse-OFF T1 | Plots: D, F |
| *13 June 2018* | SPI | Pulse-OFF T1 | Plots: E |
| | | Pulse-ON T1 | Plots: A, B, C |
| | | Tickler T0 | Plots: G, H, I |
| | Box Core | Tickler T0 | Plots: G, H, I |
| | Box Core | Tickler T0 | Plots: G, H, I |
| *15 June 2018* | Experimental Trawling | Tickler T1 | Plots: G, H, I |
| | Box Core | Tickler T1 | Plots: G, H, I |
| | | Reference T1 | R1, R2 |
| | Sonar | Tickler T1 | Plots: G, H, I |
| | Water Column | Tickler T1 | Plots: G, H, I |
| | SPI | Tickler T1 | Plots: G, H, I |



**Table 2: Results for sediment parameters for T0-T1 between Tickler, Pulse-ON, and Pulse-OFF treatments and Reference stations (mean ± SD).**

| | Tickler | | | | Pulse-ON | | | | Pulse-OFF | | | | Reference | | | |
|---|---|---|---|---|---|---|---|---|---|---|---|---|---|---|---|---|
| | T0 | n | T1 | n | T0 | n | T1 | n | T0 | n | T1 | n | T0 | n | T1 | n |
| *Sediment: very fine sand (%), chlorophyll a (µg g⁻¹), silt (%), median grain size (D50: µm)* | | | | | | | | | | | | | | | | |
| Very fine sand | 11±4.3 | 64 | **9.2±4.5** | 64 | 8.3±4.2 | 53 | 7.4±4.2 | 53 | 11±3.4 | 55 | 11±2.8 | 55 | 12±7.8 | 32 | 10±7.9 | 22 |
| Chl a | 19±31 | 64 | **8.4±17***** | 64 | 10±8.9 | 53 | **9.1±16**** | 53 | 12±16 | 55 | 19±30 | 55 | 18±18 | 32 | 14±16 | 22 |
| Silt | 13±17 | 64 | 11±14 | 64 | 8.4±6.6 | 53 | **7.7±12**** | 53 | 8.4±9.1 | 55 | 11±16 | 55 | 12±11 | 32 | 9.7±10 | 22 |
| D50 | 189±61 | 64 | 207±72 | 64 | 211±42 | 53 | 220±63 | 53 | 192±29 | 55 | 181±44 | 55 | 212±93 | 32 | 224±99 | 22 |
| *Particle mixing coefficient ($D_B$: cm² d⁻¹)* | | | | | | | | | | | | | | | | |
| $D_B$ | 0.04±0.04 | 6 | 0.22±0.26 | 6 | 0.13±0.23 | 6 | 0.17±0.16 | 6 | 0.04±0.05 | 5 | 0.12±0.24 | 5 | 0.31±0.51 | 5 | - | - |
| *Sediment profile imagery (SPI): prism penetration (mm), anoxic surface area (%)* | | | | | | | | | | | | | | | | |
| Penetration | 51±23 | 26 | 67±41 | 22 | 48±18 | 23 | 49±12 | 19 | 53±20 | 23 | 54±17 | 21 | 43±9.6 | 6 | 38±6.1 | 6 |
| Anoxic surface | 24±23 | 26 | **16±24***** | 22 | 3.8±6.1 | 23 | 5.0±10 | 19 | 22±18 | 23 | 27±28 | 21 | 0.59±1.2 | 6 | **12±13***** | 6 |

**Bold** signifies that T1 is significantly different compared to T0. *p < 0.05; **p < 0.01; ***p < 0.001.





**Table 3:** Biogeochemical results for T0-T1 between Tickler, Pulse-ON, and Pulse-OFF treatments and Reference stations (mean ± SD) for porewater nutrients (0-10 cm depth), nutrient fluxes, and mass budget model calculations. Nitrogen (N) and phosphorus (P) removal are presented as the percentage of inorganic N or P removed from the sediments after being formed from organic matter mineralization.

| | Tickler | | | | Pulse-ON | | | | Pulse-OFF | | | | Reference | | | |
|---|---|---|---|---|---|---|---|---|---|---|---|---|---|---|---|---|
| | T0 | n | T1 | n | T0 | n | T1 | n | T0 | n | T1 | n | T0 | n | T1 | n |
| *Porewater nutrients (mmol m$^{-3}$)* | | | | | | | | | | | | | | | | |
| $NH_4^+$ | 26±17 | 35 | **47±38**** | 35 | 74±59 | 34 | 81±59 | 34 | 92±83 | 36 | 80±61 | 36 | 32±25 | 18 | 23±15 | 18 |
| $NO_3^-$ | 4.1±2.7 | 35 | **6.4±4.1**** | 35 | 14±20 | 34 | 6.1±8.2 | 34 | 4.9±7.3 | 36 | **6.8±4.9**** | 36 | 7.3±7.1 | 18 | 6.0±2.8 | 18 |
| $PO_4^{3-}$ | 7.6±5.8 | 35 | 8.7±9.0 | 35 | 5.2±6.7 | 34 | **8.2±7.4**** | 34 | 9.4±15.8 | 36 | **13±9.9***** | 36 | 5.2±4.6 | 18 | 2.9±2.6 | 18 |
| $Si(OH)_4$ | 69±52 | 35 | 105±95 | 35 | 91±93 | 34 | 98±72 | 34 | 116±89 | 36 | 137±89 | 36 | 60±44 | 18 | 54±47 | 18 |
| *Nutrient Fluxes (mmol m$^{-2}$ d$^{-1}$)* | | | | | | | | | | | | | | | | |
| $O_2$ | -172±109 | 9 | **-74±40***** | 9 | -144±199 | 9 | -77±89 | 9 | -72±35 | 9 | -73±44 | 9 | -117±106 | 6 | -253±195 | 6 |
| $NH_4^+$ | 1.3±1.3 | 9 | 1.6±1.6 | 9 | 1.1±2.1 | 9 | 0.6±0.3 | 9 | 0.6±0.8 | 9 | 0.5±0.3 | 9 | 1.0±1.0 | 6 | 1.9±1.2 | 6 |
| $NO_3^-$ | 0.01±0.01 | 9 | 7e-3±0.01 | 9 | 1e-3±0.02 | 9 | 0.02±0.02 | 9 | 6e-3±0.02 | 9 | 9e-3±8e-3 | 9 | 3e-3±0.03 | 6 | -0.02±0.03 | 6 |
| $PO_4^{3-}$ | 0.03±0.02 | 9 | 0.04±0.02 | 9 | 0.01±0.03 | 9 | 0.03±0.03 | 9 | 0.05±0.07 | 9 | 0.03±0.03 | 9 | 0.03±0.04 | 6 | 0.07±0.02 | 6 |
| $Si(OH)_4$ | 1.1±0.95 | 9 | 1.8±0.76 | 9 | 0.75±0.63 | 9 | 1.1±0.90 | 9 | 1.4±1.2 | 9 | 1.1±0.70 | 9 | 0.89±0.89 | 6 | 2.4±0.68 | 6 |
| *Mass Budgets: total mineralization (mmol C m$^{-2}$d$^{-1}$), denitrification (mmol C m$^{-2}$d$^{-1}$), nitrification (mmol N m$^{-2}$d$^{-1}$), nitrogen (N) or phosphorus (P) removal (%)* | | | | | | | | | | | | | | | | |
| Mineralization | 156±98 | 9 | **68±36**** | 9 | 130±180 | 9 | 70±80 | 9 | 65±32 | 9 | 66±40 | 9 | 105±95 | 6 | **228±175*** | 6 |
| Denitrification | 28±19 | 9 | **11±6.8*** | 9 | 23±31 | 9 | 12±14 | 9 | 11±5.6 | 9 | 12±7.5 | 9 | 19±18 | 6 | 41±32 | 6 |
| Nitrification | 22±16 | 9 | **8.6±5.5*** | 9 | 18±25 | 9 | 10±12 | 9 | 9.2±4.4 | 9 | 9.5±6.0 | 9 | 15±14 | 6 | 32±25 | 6 |
| % N removal | 86±24 | 9 | 82±16 | 9 | 95±5.6 | 9 | 89±9.5 | 9 | 94±6.4 | 9 | 93±5.1 | 9 | 80±35 | 6 | 90±9.2 | 6 |
| % P removal | 96±5.1 | 9 | 92±4.6 | 9 | 99±2.6 | 9 | 92±8.1 | 9 | 93±12 | 9 | 94±5.3 | 9 | 91±15 | 6 | 92±12 | 6 |

**Bold** signifies that T1 is significantly different compared to T0. *p < 0.05; **p < 0.01; ***p < 0.001.





**Table 4: Robust regression coefficients (slope, p-value, multiple R-squared) showing the relationship of *Lanice conchilega* densities and biomass versus sedimentary faunal and biogeochemical flux parameters in trawled and untrawled sediments**

| | | PARAMETERS | Pre-trawled | | | Trawled | | |
|---|---|---|---|---|---|---|---|---|
| | | | *slope* | *p* | R$^2$ | *slope* | *p* | R$^2$ |
| *L. conchilega* DENSITIES | Sediment | Median grain size | -0.0012 | 0.29 | 0.07 | -0.0068 | 0.18 | 0.11 |
| | | % Very fine sand | **0.0002** | **0.01** | **0.33** | 0.0005 | 0.16 | 0.12 |
| | | % Silt | 0.0003 | 0.25 | 0.08 | 0.0015 | 0.14 | 0.13 |
| | Fauna | Density of other taxa | **0.460** | **<0.001** | **0.85** | 0.1030 | 0.07 | 0.05 |
| | | Species richness | **0.0003** | **0.005** | **0.27** | 0.0004 | 0.13 | 0.09 |
| | Fluxes | O$_2$ consumption | **0.021** | **<0.001** | **0.87** | 0.0020 | 0.78 | 0.01 |
| | | NO$_3$ flux | -3.2e-7 | 0.65 | 0.01 | -8.8e-8 | 0.89 | 6.9e-4 |
| *L. conchilega* BIOMASS | Sediment | Median grain size | 0.0053 | 0.78 | 0.00 | -0.163 | 0.17 | 0.11 |
| | | % Very fine sand | 0.0022 | 0.27 | 0.09 | 0.010 | 0.19 | 0.11 |
| | | % Silt | **0.0056** | **0.009** | **0.18** | 0.044 | 0.054 | 0.21 |
| | Fauna | Density of other taxa | **1.262** | **<0.001** | **0.39** | -0.571 | 0.25 | 0.01 |
| | | Species richness | **0.0166** | **<0.001** | **0.38** | 0.003 | 0.34 | 0.15 |
| | Fluxes | O$_2$ consumption | **0.427** | **<0.001** | **0.93** | 0.056 | 0.75 | 0.01 |
| | | NO$_3$ flux | **<0.0001** | **0.002** | **0.21** | <-0.0001 | 0.82 | 0.01 |

**Bolded** values show significant relationships