# Peer review of "Trawling effects on biogeochemical processes are mediated by fauna in high energy biogenic reef–inhabited coastal sediments"

_Biogeosciences, 2022_

## Author Comment (AC1)

**Response to Reviewer 1 Comments**

**Overall Response to Reviewer 1:** First we would like to thank the reviewer for taking the time to read the manuscript and give critical and constructive feedback. This study uses various types of data, and it was at times challenging to squeeze it all into a cohesive story. The reviewer gave a lot of useful suggestions to make our manuscript more concise and easier to read. This led to us removing certain superfluous aspects which were less important to the narrative. We also chose to place less emphasis on the 'reference' sites as the spatial variability of these sites make them less representative of the trawl treatment plots. All reference (now called untrawled adjacent) information can be found in the updated supplementary materials. Readers can focus on the before-after effects in the manuscript but may refer to the supplement if they are curious about potential temporal effects in the untrawled areas during the course of the experiments.

**Reviewer 1 comments**

**General comments**

This manuscript reports results from a comprehensive experimental investigation into the effects from two trawling gear types on Lanice conchilega habitats in the Vlakte van de Raan (Netherlands). Physical, biological and biogeochemical effects are well investigated and this work contributes significantly to the current evidence base, particularly regarding biochemical impacts from different gear types under varying environmental settings.

This paper matches well with the brief of the Biogeosciences journal, with other recent papers published in this journal on similar topics (e.g. Paradis, Pusceddu et al. (2019), De Borger, Tiano et al. (2021)). I would recommend the paper for publication after minor revisions.

**General concerns:**

1. **Reviewer comment:** Reference sites: Reference site 2 seems to have significantly different grain size (Fig S1). Reference sites also have large variation in O2 consumption and species densities (Fig 6), as well as higher mean mixing rates (Table 2) and generally higher species densities when compared to treatment sites (Fig 6). Are these sites definitely representative of treatment sites? Information should be provided in the methods on how it was assured that all treatment sites and reference sites were generally environmentally representative of each other.
   **Response:** This is a valid issue that both reviewers have highlighted. The area had so much spatial variability that the 'reference' sites were not representative of all the treatment areas. The locations of the reference sites were, however, directly adjacent to the treatment sites. Data from these sites have thus been separated and named 'Adjacent 1' (AD1), and 'Adjacent 2' (AD2). We have also, now incorporated the third untrawled adjacent site (AD3) into text and Fig. 1 and have specified that the majority of its data came only at the T1 timestep (end of experimental period). When comparing between T0-T1 trawled treatments, we did not incorporate the "reference" sites in the analysis as we thought a before-after comparison would be more appropriate due to the large differences in spatial variation. Although inclusion of the reference/adjacent sites may help qualitative comparisons (observing potential temporal effects), to keep readers focused on the pure trawling effects, we have decided to move the reference/adjacent results from the main figures/Tables in the manuscript to the supplementary material. Upon further inspection, we also found a small error in the reporting and have seen that some of the reference T1 data was missing so this has been corrected. All reference/adjacent results can now be found in Supplementary Table S5. We have also removed the text referring to the BACI design of this study as perhaps it is not appropriate due to the before-after statistical comparison (i.e. lacking control).

2. **Reviewer comment:** Methods: It is not always clear how data analyses were undertaken e.g. Were the non-parametric tests undertaken in R and with what packages. How were the biological mixing models on bioturbation analysed, and with what software? Make sure sufficient detail is provided throughout the methods where necessary.
   **Response:** We decided to exclude the results from the bio-mixing models as we are not confident that our results ($D_B$) effectively describe the details of sediment transport in our study. As much of the data were consistent with non-local mixing models (often the case when tubeworms eat OM at the surface and inject it in depth) the $D_B$ values only describe part of the sediment transport.

We have moved and modified our sentence stating that all statistics were conducted (and figures were created) using R, to the beginning of the statistics subsection in the methods. We also specify the R package used to carry out the mass budget analysis ("limSolve"; Soetaert et al., 2009) in the main text of the manuscript but we still feel like the detailed description of this methodology is best located in the supplementary material (our methods section is quite long already). Other software such as SpiArcBase (SPI), Qimera (bathymetry) and FMGeocoder Toolbox (backscatter) are specified in their respective sections in the methodology.

3.  **Reviewer comment:** Results: Some of the results seem superfluous. e.g SMP POC and TN; and backscatter testing between all pairwise combinations and factors. Throughout the manuscript, results should only be reported where they contribute to the narrative of the paper and are mentioned within the discussion. I realise it is tempting to report all results which were produced within the investigation but, results should be removed where they are superfluous to the narrative, discussions or conclusions.
    **Response:** We have removed the water column results for SPM, POC and TN and have also removed the backscatter results describing potential differences between sample sites. The between site comparisons were meant to give some indication of the spatial variability of the experimental area, however, we see that this may not be communicated well and is probably confusing for readers. We have, thus moved the backscatter figure (originally Fig. 3) to the supplementary material (Fig. S6).

**Specific comments**

4.  **Reviewer comment:** L16: To add context it should be noted within the abstract that this is looking at "acute impacts by experimental disturbance from representative fishing gears"; or other similar phrasing conveying the same message.
    **Response:** The text has been edited to show that the two types of bottom trawls are specifically used to exploit North Sea sole.
    > "This study explores the physical, biological and biogeochemical effects of  bottom trawl fishing on a coastal ecosystem dominated by the tube building polychaete, *Lanice conchilega*. Two specific gears types, both used to exploit North Sea sole (*Solea solea*), were compared: electric pulse trawls and tickler chain rigged beam trawls."

5.  **Reviewer comment:** L23: Taxon densities and species richness of what?
    **Response:** We have edited the text to state "macrobenthos densities and species richness".

6.  **Reviewer comment:** L35-36: It seems contradictory to say L. conchilega habitats are under "considerable threat" from bottom trawling, and then state that they have "biological resilience to bottom trawl impacts".
    **Response:** Our original text here was not so clear. We want to state that biogenic reefs are well known to be vulnerable to trawl disturbance (Fariñas-Franco et al., 2018; Grabowski et al., 2014; Kaiser et al., 2002), however, *L. conchilega* habitats have been linked with resilience to bottom trawling (Rabaut et al., 2008). We added a few key words to these sentences which will hopefully help convey this point:
    > "Bottom trawling activities are a common occurrence in North Sea *L. conchilega* dominated habitats (Rabaut, 2009) and can present a considerable threat to most biogenic reef ecosystems (Kaiser et al., 2002; Fariñas-Franco et al., 2018). Biological resilience to bottom trawl impacts have, however, been linked to both *L. conchilega* reefs (Rabaut et al., 2008) and the dynamic sandy habitats they are often found in…"

7.  **Reviewer comment:** L40: Alter to state "Bottom fishing affects carbon cycling on the seafloor" or "in seabed sediments".
    **Response:** We have modified the text to state "benthic carbon cycling".

8.  **Reviewer comment:** L42-43: again add "reduced mineralization of organic carbon in seabed sediments".
    **Response:** We have added, "in seabed sediments" to the end of the sentence.

9.  **Reviewer comment:** L91: State the minimum distance experimental areas were separated by.
    **Response:** The treatment plots were separated by a minimum of 100 m to try to avoid indirect impacts from nearby trawling. This information has now been added to this section.

10. **Reviewer comment:** L92-93/Fig 1: It would be clearer if these three sampling stations (western, middle and eastern) were shown within the figure. E.g. with lined boxes.
**Response:** We attempted to create the map using lined boxes as suggested, however, this crowded the image and made it difficult to view the sample locations. The significance of the northwest, central and southeast stations (changed from: western, middle and eastern) was simply to gather three replicates representative of each trawled plot.

11. **Reviewer comment:** L94: What was the size of the reference areas? How close were they to treatment areas?
**Response:** The reference areas were within 200-250 m to the nearest trawling locations as seen in figure 1. For this, we simply specified a point on the map which was near the trawled areas but far enough away to have a low chance of disturbance. The actual sampled locations varied within 80 m of the specified sampling coordinates. The text has been edited to describe these points.

12. **Reviewer comment:** L94: It would be good to give some descriptive information on how the different treatment and reference areas were determined to have similar/same environmental and biological settings.
**Response:** The orientation and location of treatment plots and reference/adjacent areas were determined based on their position in a particular bathymetric depression. This depression was chosen based on previous data showing high macrobenthic biomass coupled with low trawling effort (as described in the first paragraph of the methods). We have added information regarding the position of the treatment plots and reference/adjacent areas to show that they were position to represent the different sections within this bathymetric depression.

13. **Reviewer comment:** L102: It would be clearer to move Table 1 to the supplementary material as it is overly detailed - simply put a descriptive sentence here (L102). E.g. that samples were collected between x-x hours before trawling and x-x hours after.
**Response:** This issue was also pointed out by Reviewer 2. The table specifying exact sampling details has been moved to the supplementary materials. We have also shortened the text and referred to the table specifying sampling dates (Table S1).

14. **Reviewer comment:** L156-59: Here, and in other places within the methods (e.g. L141-146), you do not need to repeat the dates of sample collection if they are already reported in Table 1. If you want to add a brief descriptor in terms of time before/after trawling then try to be more general as suggested above (with reference to Table 1 for more detail).
**Response:** The text has been shortened and modified to refer to Table S1 in the supplement which contains more detailed sampling information.

15. **Reviewer comment:** L161: I think it would be clearer if this simply said "Two out of the three box cores were subsampled for sediment parameters from each treatment and reference station".
**Response:** We have edited the text, as suggested, by the reviewer (it sounds nicer).

16. **Reviewer comment:** L163: Did all of the sediment pass through 1mm? If not, was this fraction considered?
**Response:** Various shells and shell fragments did not pass through the 1mm sieve, however, these larger particles were not considered for grain size analysis. As seen in supplementary Fig. S1 most sediment particles measured were well under 500 micrometres.

17. **Reviewer comment:** L179: Should be simplified to state "Pore water was collected from the same box cores as used for sediment parameter sampling".
**Response:** As suggested, the text has been modified to show that the porewater cores were collected in the same box cores as the sediment parameter samples.

18. **Reviewer comment:** L293-295: The details of the results from all the different bio-mixing models do not seem to be reported. They should be present in a Supplementary table with the AIC scores for each model and the selected optimal model highlighted.
**Response:** As detailed in the response to comment 1, we have removed information regarding the bio-mixing models. These were a late addition to the study and although, we were excited to include them, they probably do not fit the narrative so well.

19. **Reviewer comment:** L230: LMMs were used to investigate what about the stated responses variables? Significant differences between T0 and T1 for each treatment separately? This needs to be explained better.
    **Response:** We have modified the text to clarify that we are making statistical comparisons from before fishing (T0) and after fishing (T1) for individual parameters and fishing treatments. See text as follows:
    "Linear mixed effects models (LMM) were used to investigate significant differences before and after fishing (T0 – T1) for:  SPI measurements, sediment parameters, porewater nutrient concentrations, biogeochemical flux, mass budget model results and ecological characteristics ( macrobenthos densities, biomass, species richness) using the *lmer*-function in the R package: "lme4" (Bates et al., 2015). For  a given variable and treatment (example: oxygen flux and tickler treatment), a "model" was specified using the "timestep" (T0  - T1) as a fixed effect variable, "temperature" as a co-variate, and "station" as a random effect variable . A "reduced-model b" was created to consider only the random effect variable (station). The full  and reduced models were tested against each other using a partial F test. This approach was taken to assess the effect of fishing disturbance (timestep = before-after statistical comparison) in respect to temperature while minimizing spatial autocorrelation between stations."

20. **Reviewer comment:** L250-253: There is no use/discussion of SPM POC and TN in the discussion. Remove from results as the information seems to be superfluous.
    **Response:** These results have now been removed. There was some discussion about these topics in previous versions of the manuscript, however, since that has been removed, it is also appropriate to remove these parameters from the results (and methods).

21. **Reviewer comment:** L269-270: Backscatter results seem overly explorative as much of the reported differences (or lack of differences) are not mentioned in the discussion. E.g. Why are you testing for differences between sites separate from the consideration of trawling impact?
    **Response:** The inclusion of the backscatter data in the different plots (as well as between T0-T1 fishing) was meant to convey the large amount of spatial variation. We understand, however, that this is not so clear and decided to move the backscatter figure to the supplementary material.

22. **Reviewer comment:** L265: It is not clear where these statistical tests have been reported. The legends of Table S1 and Table S2 should be expanded so it is clear how statistical significance was determined.
    **Response:** Some text has been added outside of the table legends to describe the statistical results in these tables. We felt that this information would be better communicated outside of the table legends.

23. **Reviewer comment:** L268-269: Again Fig 3 does not report the statistical tests. Where are these reported?
    **Response:** We originally reported that there no significant p-values with the Benjamini-Hochberg correction (Dunn test) between experimental plots. We then reported uncorrected p-values showing significant differences between certain plots. As corrected p-values show no significant differences we think we should leave the figure unchanged but have decided to move it to the supplementary material as it describes a minor part our study. Hopefully this helps streamline the reading of the manuscript. We have also removed the sentences regarding uncorrected and corrected statistical differences between experimental plots to have readers focus more on the trawling effects.

24. **Reviewer comment:** L270: Again, where are these p-values and pairwise differences reported? They are not shown in Fig. 3
    **Response:** As described in the response to comment 23, we have removed the sentences regarding uncorrected p-values.

25. **Reviewer comment:** L284-287: It is not clear what this adds to the story of the paper. Remove to aid clarity.
    **Response:** This text has been removed to streamline the reading.

26. **Reviewer comment:** L287: In the methods "Robust regression analysis" is only discussed in relation to Section 2.7.4 (Impact of Lanice conchilega). It is not clear why/how it is used here.
    **Response:** This sentence is not so important to the narrative (and may also be superfluous knowing the relationship between chl-a and silt) so it has been removed.

27. **Reviewer comment:** L288-290: Add "between T0 and T1" to the end of this sentence to aid clarity.
    **Response:** The text has been edited to make it clear that it is referring to T0 – T1 fishing treatments.

28. **Reviewer comment:** L299-303: There is a lot in the results, to keep the focus of the reader I would remove this more descriptive information if it is not used later in this paper. Its presence in the supplementary is sufficient.
**Response:** The descriptive information that does not refer to statistical significance, has been removed. A sentence has been added to refer readers to Table 1 and Table S5 for more information on nutrient fluxes.

29. **Reviewer comment:** L311: Restructure sentence so it starts "For all fishing treatments….". Just so it is clear you are no longer talking about just Tickler.
**Response:** We have modified the sentence to begin with the suggested phrase.

30. **Reviewer comment:** L343: It is not clear why Fig 3 is specifically referenced here.
**Response:** The figure describing backscatter differences was also meant to convey spatial variability. This is, however, not so intuitive so the reference has been removed.

31. **Reviewer comment:** L347-349: Remove this sentence. There is no need to introduce the discussion by saying you are going to "discuss".
**Response:** This sentence has been removed.

32. **Reviewer comment:** L360-369: Is it not also possible that increased current speeds would transport trawl-induced resuspended sediment away from potential sampling equipment more quickly, therefore reducing likely SPM effect? Or is it considered that the sampling equipment were too close to the trawl disturbance for this to be an influencing factor?
**Response:** As described in section 2.3 (Water Column measurements [methods]) the RV positioned itself within 80 m of the trawled area during trawling while making sure to be positioned against the current and the trawl disturbance to (hopefully) capture the effect of the sediment cloud. It is difficult to say whether this was close enough to capture the trawling disturbance as we could not visibly observe the extent of the sediment plume caused by the trawling. Using the same method, we were able to clearly record the effect of trawl-induced sediment resuspension in the Frisian Front with an instrument located further away (100 m) with lower current speeds (Tiano et al., 2019). The sediments in the current study, however, are sandier than those in the Frisian Front and so it may be that the resuspended sediments settled much faster before they could be detected. It may also be possible, as mentioned by the reviewer, that the increased current speeds could have hindered the detection of trawl effects, though we speculate that this may be more of a dilution effect from increased dispersion of particles.

    We have added some text speculating on dilution based on current-induced dispersion of resuspended particles in high energy habitats.

33. **Reviewer comment:** L376: Remove statement about variation between northern and southern plots. It does not add anything to the messaging of the paper.
**Response:** This statement has been removed.

34. **Reviewer comment:** L389: Some mention should be made into the significant difference in anoxic sediment in the reference sites between T0 and T1, and how this may influence results of treatment sites.
**Response:** We have modified the text here to include the anoxic surface area results although after separating the reference/adjacent sites, this was only statistically significant for one of the sites. We have, thus specified this and ensured that the text is more speculative rather than indicative.
    "Furthermore, higher percentages in the anoxic surface area from the SPI images taken in adjacent sites (significant for AD1) may suggest increased mineralization in undisturbed areas during the course of the experiments (Table S5 in the Supplement). Mass budget estimates for mineralization were also higher on average at T1 for AD1 and AD2, though this was not statistically significant. If the benthic metabolism in untrawled areas did increase during the experimental period, the trawling effects detected in this study (lower metabolism after trawling) may be considered to be conservative estimates"

35. **Reviewer comment:** L394: Remove discussion of reference sites – again it is not clear how this contributes to the discussion and messaging of the paper in general.
**Response:** This text has been removed.

36. **Reviewer comment:** L396: I think it is important to reiterate here that it was not statistically significant.
**Response:** The text regarding sediment mixing has been removed as detailed in the response to comment 2.

37. **Reviewer comment:** L406-407: This is already mentioned in Section 4.2; no need to repeat here.
    **Response:** The text has been removed to reduce redundancy.

38. **Reviewer comment:** L408: Change to "Despite the lower impact of Chl a"
    **Response:** This has been changed to "Despite the lower trawl penetration into the sediment" to provide clarity. There was only a relatively lower impact on Chl a compared to Tiano et al., (2019), however, the absolute levels of Chl a removed in the current study were much higher.

39. **Reviewer comment:** L471-472: Rephrase, to remove "environmental benefit". Fishing cannot really be said to have environmental benefit. Perhaps change to say "there is potential for reduced environmental impact from pulse trawls due to its higher catch efficiency……".
    **Response:** We changed the wording of this sentence to remove the phrase "environmental benefit" and emphasize the potential for a reduced environmental footprint stemming from increased catch efficiencies rather than any decrease in direct impacts.

40. **Reviewer comment:** L745: Alter to state "Example sediment profile imagery…"
    **Response:** The text has been edited.

41. **Reviewer comment:** Tables: Table 2 and Table 3 should be combined. This would aid clarity and there is no need for them to be separated.
    **Response:** These tables have now been merged. The newly formed table in the manuscript is now Table 1 (the table with sampling information was moved to the supplement).

**Technical corrections**

42. **Reviewer comment:** L134: I assume this is meant to state T1 rather than T2.
    **Response:** This sentence is probably not necessary and has been removed.

**Literature Cited**

Fariñas-Franco, J. M., Allcock, A. L., & Roberts, D. (2018). Protection alone may not promote natural recovery of biogenic habitats of high biodiversity damaged by mobile fishing gears. *Marine Environmental Research*, *135*(September 2017), 18–28. https://doi.org/10.1016/j.marenvres.2018.01.009

Grabowski, J. H., Bachman, M., Demarest, C., Eayrs, S., Harris, B. P., Malkoski, V., Packer, D., & Stevenson, D. (2014). Assessing the Vulnerability of Marine Benthos to Fishing Gear Impacts. *Reviews in Fisheries Science & Aquaculture*, *22*(2), 142–155. https://doi.org/10.1080/10641262.2013.846292

Kaiser, M. J., Collie, J. S., Hall, S. J., Jennings, S., & Poiner, I. R. (2002). Modification of marine habitats by trawling activities: prognosis and solutions. *Fish and Fisheries*, *3*, 114–136.

Rabaut, M., Braeckman, U., Hendrickx, F., Vincx, M., & Degraer, S. (2008). Experimental beam-trawling in Lanice conchilega reefs: Impact on the associated fauna. *Fisheries Research*, *90*(1–3), 209–216. https://doi.org/10.1016/j.fishres.2007.10.009

Tiano, J. C., Witbaard, R., Bergman, M. J. N., Rijswijk, P. Van, Tramper, A., Oevelen, D. Van, & Soetaert, K. (2019). Acute impacts of bottom trawl gears on benthic metabolism and nutrient cycling. *ICES Journal of Marine Science*. https://doi.org/10.1093/icesjms/fsz027

---

## Author Comment (AC2)

**Response to Reviewer 2 Comments**

**Overall Response to Reviewer 2:** We thank the reviewer for the constructive comments on the manuscript as well as the time taken to read through and give a thorough evaluation. Perhaps the largest change to our manuscript was the exclusion of the bio-mixing model results due, in large, to the feedback from the reviewer and further reflection on the topic. We realize that in these sediments, the reported $D_B$ values is probably too simplistic to properly describe mixing due to the degree of non-local particle transport (facilitated by *L. conchilega*). These results were a late addition to the study and did not contribute greatly to the story and by removing it we also make our study more concise.

**Reviewer 2 comments**

**General comments**

This contribution presents results from an intense in situ experiment, investigating into trawling effects inflicted by different trawling gears on a shallow, 10 m water depth, sandy sediment inhabited by the polychaete Lanice conchilega, which locally forms dense tube "lawns", here termed biogenetic "reefs". The investigation encompasses physical impact on sediment structure, biological and biogeochemical effects thereof. It is well conducted and the data are by enlarge well interpreted; thus, this study is an important contribution to literature. The shallow water setting in particular, the different disciplines involved as well as a substantial data set on biogeochemical aspects of the impact, will make it a valuable publication.

It will contribute both to publications in Biogeosciences and in other journals on this growing body of evidence. I recommend the manuscript for publication subject to minor revisions.

**Major points**

1. **Reviewer comment:** Bioturbation results …

    Did you average $D_B$ from different models? This would involve averaging $D_B$ representative of different portions of the overall particle transport ($D_B$ represents all of transport in model 2, however only parts of the overall transport in higher models)

    Please comment on this and add information. This becomes particularly important in the context of results discussed in line 296, i.e. "physical trawling enhances $D_B$".
    **Response:** The reviewer highlights some important issues regarding the bio-mixing models used in this study. The reviewer is correct in that the $D_B$ values were averaged from the best fitting models, however, as stated by the reviewer, particle transport as represented by the $D_B$ value only describes some of the dynamics in the bio-mixing models for non-local exchange. *L. conchilega* often caused Chl-a to be injected in depth so non-local mixing models were often used to estimate sediment mixing. It may be too simplistic in this situation (with high non-local exchange) to generalize these $D_B$ values in relation to trawling for this study. We also only have limited evidence (non-significant results) of trawling potentially increasing sediment mixing rates. This, and the fact that the reading of this study may be hindered by having too much information, has led us to remove the sections of the manuscript regarding sediment mixing as described by the bio-mixing models. We hope that this will streamline the reading and help readers focus on the main results of this manuscript.

2. **Reviewer comment:** SCOC …

    Could you please comment on the unusually high absolute values (see also recent database by Stratmann et al. 2022)?

    SCOC of >400 mmol $m^{-2}$ $d^{-1}$ at ~500 g WW $m^{-2}$ conchilega or ~700 mmol $m^{-2}$ $d^{-1}$ at ~1400 g WW $m^{-2}$ seem at least questionable! My reasoning is as follows. The later biomass could be responsible for around 100 mmol oxygen $m^{-2}$ $d^{-1}$, if biomass-specific respiration is around 2.5 mg $O_2$ $g^{-1}$ $h^{-1}$ as for smaller sized polychaetes (e.g. Bennett and Rakocinski 2020). However, who or which additional process in a lawn of Lanice conchilega could respire an additional 600 mmol $m^{-2}$ $d^{-1}$?

**Response:**

[Figure]

[Figure]

We were also surprised at the very high SCOC values measured in this study. We are, however, quite confident in our methodology and measurements. Above you can see some of the raw data from a core which measured an $O_2$ flux of 663 mmol m$^{-2}$ d$^{-1}$ (the highest flux reported in this study) compared (underneath) to with varied oxygen flux estimates that represent more "normal" values. You can see in the first image that almost 100 mmol of oxygen was consumed within 30 minutes in a core with 15 cm of overlying water (the core was subsequently aerated). While this $O_2$ consumption is exceptionally high, biomass-specific respiration from *L. conchilega* is only one of several factors which can lead to the high consumption of oxygen in these sediments.

- High summer temperatures: During the experiments (and the week leading up to it), we experienced particularly warm temperatures (for early June in Belgium/Netherlands) with surface waters fluctuating between 16-17 degrees °C. This probably stimulated benthic metabolism and macrobenthos (and microbial) activity.
- Bioirrigation: fauna-mediated ventilation of the sediments is well known to enhance microbial respiration by greatly expanding the surface area available for microbial respiration and other oxidation processes (Kristensen and Kostka, 2013). The SPI images in our study provide evidence of strong bioirrigation with high levels of oxidized sediment when tubeworms were present.
- Microbial respiration, which can easily exceed 90 % of benthic $O_2$ uptake (Glud, 2008), is almost certainly significant factor in the high $O_2$ consumption found in our study.
- Anoxic mineralization: In addition to oxic respiration, $O_2$ is also taken up by the oxidation of previously reduced substances (Glud, 2008). Substances produced through anoxic mineralization diffuse towards the sediment surface and consume oxygen once reaching oxic conditions (Soetaert et al., 1996).
- Suboxic reactions: Oxygen is consumed with reactions from nitrification and iron reduction.

[Figure]

- Respiration and activity from other benthos: These sediments often contained high biomass from invertebrates other than *L. conchilega* and also contributed to oxygen consumption from direct respiration, bioirrigation and bioturbation (which can mix OM into deeper sediment layers and can enhance mineralization).
- High OM influx (Chl-a): Perhaps most importantly, these sediments exhibited very high Chl-a values. To put this in context, the average concentration of Chl-a in the T0 Tickler plots was 19 $\mu$g g$^{-1}$ which was associated with an average $O_2$ consumption of 172 mmol m$^{-2}$ d$^{-1}$ while productive sandy mud habitats in the central North Sea can average 3 $\mu$g Chl-a g$^{-1}$ linked with $O_2$ consumption values of 18 mmol m$^{-2}$ d$^{-2}$ (Tiano et al., 2019). The high input of fresh organic material (its degradation increasing the $O_2$ flux) coupled with high bioirrigation caused by *L. conchilega* is the most likely culprit causing the exceptionally high $O_2$ consumption from the experimental site.

De Smet et al., (2016) documented *L. conchilega* communities respiring, on average, 193 and 99 mmol C m$^{-2}$ d$^{-1}$ in two different intertidal locations in France (C and $O_2$ fluxes are comparable). This is within the range of the average $O_2$ consumption values from most of the samples within this study (Table 1 in the updated manuscript).

Maximum North Sea summer temperatures can typically peak around 18 °C in August (and only in southern coastal waters). We have added some information about the measured water temperatures in the results section describing water column conditions to describe the elevated (for June) temperatures:

> "During the experimental period, the water column exhibited mixed thermal conditions with temperatures ranging between 15.8 – 17.0°C."

3. **Reviewer comment:** The reference stations are not ideal in that they represent extremes in some measures and do not represent an expected average background (high oxygen demand, high faunal density, intense bioturbation). It is necessary to address this issue for it obviously raises the questions if the trawling effects can be and are at all compared to the references, or if they are only compared between T0 and T1 on trawling plots. Does the statistical analyses take care of this? It is hard to see this easily.
   **Response:** Both reviewers have highlighted the issue of reference stations not being completely representative of the trawled locations. Indeed, the spatial variability in the reference sites make it difficult to argue that they represent the same conditions in the trawled areas (which also have a high degree of spatial variability). Because of this, we have decided to move the information from the reference sites to the Table S5 in the supplement and have separated the results for each particular site (previously, R1 and R2 results were averaged). We have also included information on the 3$^{rd}$ reference area but specify that most of the information came during the T1 timestep (the end of the experimental period). The reference sites have been renamed the "untrawled adjacent sites" (AD1, AD2, AD3). These were not accounted for when statistically assessing trawl impacts, therefore, we removed BACI references in the text as our study only makes before-after statistical comparisons which are more appropriate in this situation. Upon close inspection, we also found an error in the reference/adjacent analysis and have some missing data for some grain size in the reference/adjacent T1 timestep and have corrected this this. If readers are curious about potential temporal effects or any other parameters in the untrawled adjacent areas, they can refer to Table S5 in the supplement.

**Specific comments**

4. **Reviewer comment:** L 41: "maximize" does not seem the right word here. Alterations of sediment structure implies changes of (possibly steady-state) diagenetic conditions. "Impairing biogeochemical processes"?
   **Response:** This text refers mainly to Ferguson et al., (2020)'s results and discussion of trawling disrupting the suboxic areas in the sediment matrix where denitrification can be maximized. We have edited the text as suggested.

> "Bottom fishing affects benthic carbon cycling by displacing bottom dwelling organisms (Hiddink et al., 2017; Sciberras et al., 2018),  altering sediment structures  leading to the impairment of biogeochemical processes…"

5. **Reviewer comment:** L 61: the sentence would be more straightforward if it read: "With respect to sensitivity to direct impact and recovery potential, coarse sediment ecosystems characterized by high levels of natural disturbance typically display more resistance and resilience to bottom fishing …"
   **Response:** We have taken the reviewers suggestion (and nicely worded sentence) and incorporated this into the text.

6. **Reviewer comment:** Fig 1: The insert dimensions are MUCH smaller (~1.8 km wide) than the square indicating the location on the left map in Figure 1 (~20 km wide). This is a bit misleading and could be changed by reducing the square's size.
   **Response:** The square in the left part of the figure (large map) is now smaller to better describe the dimensions of the experimental area.

7. **Reviewer comment:** Tab 1: This table could easily move to supplements. It contains only background information that is not necessary for understanding the main text.
   **Response:** This table has been moved to the supplementary material (now supplementary Table S1).

8. **Reviewer comment:** L 125: "wide" should be "width"
   **Response:** This has been changed to "width".

9. **Reviewer comment:** L 154: The total number of 69 box cores cannot be understood without the information that at R3 only T1 was sampled! A total would be 72 (2 times 3 replicates at 9+3 plots/reference station, i.e. 6 x 12=72). Not sampling R3 for T0 reduces this number by 3. Right? Reword to include that R3 only T1 was sampled.
   **Response:** The total number of box cores can be calculated by multiplying the treatment plots (9) and the first two adjacent stations by 3 replicates and 2 timesteps $((9 + 2) * 3 * 2 = 66)$. The three cores from the third adjacent site leads to 69 total box cores taken. The text has been edited to show the inclusion of AD3 (formerly R3) benthic samples only at the T1 timestep and AD3 has been added to the map figure.

10. **Reviewer comment:** L 171-177: Following these details describing the model "family" there is no information in lines 193 and thereafter as to why only $D_B$ is reported.
    **Response:** As detailed in the response to comment 1, we have removed the portions of the manuscript which refer to the bio-mixing model results.

    Was there no non-local effect visible in the modeling results? Maybe report (some) of these results as well in the supplement. And consider the information in $D_B$ as mentioned in the major points above.
    **Response:** As described in the response to comment 1, there were many cores where non-local mixing was observed which complicated matters when trying to assess and compare trawl-induced sediment mixing with natural bioturbation. Ultimately, these results do not provide much robust insight (not statistically significant) on the effects of trawling on sediment mixing so decided to remove the sections of the manuscript where the bio-mixing models were used.

11. **Reviewer comment:** L 187: incubation on board or at land, how much time to settle after coring impact?
    **Response:** This paragraph has been restructured and edited to describe the incubations occurring inside the research vessel after settling for ~ 6 hours.

12. **Reviewer comment:** 2.7.3. (SPI and benthic sample analysis) after reading this and not acquainted with the specific analyses, my impression is that this allows to separate effects of T0 or T1 and temperature at the same station for the mentioned parameters? Add some information explaining what the described procedure yields, please.
    **Response:** This paragraph has been reworded to provide more clarity in our analysis and also why we chose this approach. This analysis just assesses statistical differences between T0-T1 while correcting for temperature and accounting for spatial variation between sample sites.

    "Linear mixed effects models (LMM) were used to investigate significant differences before and after fishing (T0 – T1) for: SPI measurements, sediment parameters, porewater nutrient concentrations, biogeochemical flux, mass budget model results and ecological characteristics ( macrobenthos densities, biomass, species richness) using the *lmer*-function in the R package: "lme4" (Bates et al., 2015). For  a given variable and treatment (example: oxygen flux and tickler treatment), a "model" was specified using the "timestep" (T0  – T1) as a fixed effect variable, "temperature" as a co-variate, and "station" as a random effect variable . A "reduced-model" was created to consider only the random effect variable (station). The full Modeland reduced models were tested against each other using a partial F test. This approach was

taken to assess the effect of fishing disturbance (timestep = before-after statistical comparison) in respect to temperature while minimizing spatial autocorrelation between stations."

13. **Reviewer comment:** Tables S1 and S2 have much too little information their respective legends. Please add.
    **Response:** We have added text describing the results and statistics outside the legend for these tables. We think that this information will be better communicated this way instead of inside the legend (especially the descriptive results).

14. **Reviewer comment:** L 268-270: if all these statistical results are provided with reference to Figure 3, they should be shown there. I cannot find that statistical information in Fig 3!
    **Response:** We originally reported corrected and uncorrected (Dunn test with a Benjamin-Hochberg correction) p-values to illustrate spatial differences in backscatter. This was not communicated very well and possibly takes away from the narrative so these sentences describing statistics between plots have been removed for readers to focus on the fishing effects. As we have now decided to only use Benjamin-Hochberg corrected p-values values (and only for fishing effects), this leaves no statistical differences in the plot to be reported. We also decided to move this plot into the supplementary material (Fig. S6) as the information does not contribute strongly to the story.

15. **Reviewer comment:** L 280 and thereafter: Some of the results reported here are somewhat superfluous for they display commonly known relations of sediment grain size and other sediment related parameters (coarser grain size is associated with less fines, more fine material usually correlates with more chlorophyll). Thus, the passage could be a bit shorter.
    **Response:** We removed much of the superfluous text describing a significant increase towards larger grainsize as there is a text describing the decrease in silt fractions. We also removed text regarding the relationship between Chl-a and silt and have combined/shortened the sentences describing the particle size distribution curves (Fig. 4)

16. **Reviewer comment:** Figures 6 and 7: why is SCOC (fig.7) called oxygen consumption in Fig 6? Are those not the same values as in Fig 7?
    **Response:** We thank the reviewer for pointing this out and have changed '$O_2$ consumption' to 'SCOC' to remain.n consistent with the terminology.

17. **Reviewer comment:** L 348: the passage "to physical, biogeochemical, and ecological characteristics" ios again mentioned in line 357 as "simultaneously investigates acute beam trawl impacts on biological (Rabaut et al., 2008), physical (Depestele et al., 2016) and biogeochemical dynamics". I suggest removing it in line 348.
    **Response:** We have removed this sentence as suggested.

18. **Reviewer comment:** L 375 and thereafter: This is a lot of text for little change seen or measured! Can it be reduced?
    **Response:** The text here has been edited to make it more concise.

19. **Reviewer comment:** L 396: Did $D_B$ really increase? It may well be so, however, an increased burial with more random mixing of sediments is not always the case. Increased mineralization of labile material may counteract overall burial. A more cautious wording, such as "it may result in altered OM diagenesis and nutrient cycling", is warranted.
    **Response:** As described in the response to comment 1, we have removed the bio-mixing results. These data were a late addition to the study and while we were initially excited to include them, we are less confident in these than the other results.

20. **Reviewer comment:** L 406: this is an unnecessary repletion of the bathymetry discussion above
    **Response:** This sentence has been removed.

21. **Reviewer comment:** L 420: Lanice additionally extends its tubes up to 20 cm below the surface. Therefore the subsequent sentence (line 22) sentence should state "… to organisms without pronounced protruding sediment surface structures."
    **Response:** We have restructured and edited the sentences as suggested.

"These results also suggest that areas with *L. conchilega* may be vulnerable to relatively shallow (~ 1 cm) seabed disturbances as their tubes extend from over 10 cm within the sediment to above the  seabed surface. Contrary to this, faunal-mediated biogeochemical functions in the Frisian Front are  relegated to organisms such as the burrowing mud shrimp, *Callianassa subterranea*, that do not exhibit any protruding sediment surface structures and resid deeper within the  seabed (Amaro et al., 2007; Tiano et al.,

2020).  These organisms are probably less affected by trawling despite their habitats displaying  softer sediments which are prone to greater trawl penetration (Depestele et al., 2019; Tiano et al., 2019, 2020)."

22. **Reviewer comment:** L 464: This is a bit of discussion and maybe should not be in conclusions.
    **Response:** We have removed the sentence discussing different results in the literature between pulse and beam trawls to keep the focus more on the conclusions of the manuscript.

23. **Reviewer comment:** L 479: "removal of sedimentary carbon"; this is why above there should be no statement of enhanced D'B' increasing burial!
    **Response:** We think that the effects of trawling are more complicated than just the removal of carbon from the sediment surface. While this trawl induced erosion and resuspension of OM is something that we have observed several times in the field, trawling also mixes benthic sediments underneath the eroded surface layer (Depestele et al., 2018), and has been hypothesized to mix OM into deeper layers potentially resulting in increased carbon burial (Mayer et al., 1991). The net result of this is still a mystery as we do not yet know the extent of the trawl-induced mixing nor the fate of the resuspended carbon (how much is mineralized in the water column/how much is deposited in other areas). Either way, with the removal of the bio-mixing portion of the manuscript, our statement about enhanced burial has been removed.

    **Technical corrections**

24. **Reviewer comment:** L 525 "(Blackburn 1988)" does not belong in this Breakman et al. 2010 citation
    **Response:** We thank the reviewer for pointing out this typo and have corrected it.

**Literature Cited**

De Smet, B., Van Oevelen, D., Vincx, M., Vanaverbeke, J., & Soetaert, K. (2016). Lanice conchilega structures carbon flows in soft-bottom intertidal areas. *Marine Ecology Progress Series*, *552*(June), 47–60. https://doi.org/10.3354/meps11747

Depestele, J., Degrendele, K., Esmaeili, M., Ivanovi, A., Kro, S., Neill, F. G. O., Parker, R., Polet, H., Roche, M., Teal, L. R., Vanelslander, B., & Rijnsdorp, A. D. (2018). Comparison of mechanical disturbance in soft sediments due to tickler-chain SumWing trawl vs . electro-fitted PulseWing trawl. *ICES Journal of Marine Science*, *fsy124*, 1–18. https://doi.org/10.1093/icesjms/fsy124

Ferguson, A. J. P., Oakes, J., & Eyre, B. D. (2020). Bottom trawling reduces benthic denitrification and has the potential to influence the global nitrogen cycle. *Limnology and Oceanography*. https://doi.org/10.1002/lol2.10150

Glud, R. N. (2008). Oxygen dynamics of marine sediments. *Marine Biology Research*, *4*(4), 243–289. https://doi.org/10.1080/17451000801888726

Kristensen, E., & Kostka, J. E. (2013). Macrofaunal Burrows and Irrigation in Marine Sediment: Microbiological and Biogeochemical Interactions. *Interactions Between Macro- and Microorganisms in Marine Sediments*, 125–157. https://doi.org/10.1029/CE060p0125

Mayer, L. M., Schick, D. F., Findlay, R. H., & Rice, D. L. (1991). Effects of commercial dragging on sedimentary organic matter. *Marine Environmental Research*, *31*(4), 249–261. https://doi.org/10.1016/0141-1136(91)90015-Z

Soetaert, K., Herman, P. M. J., & Middelburg, J. J. (1996). A model of early diagenetic processes from the shelf to abyssal depths. *Geochimica et Cosmochimica Acta*, *60*(6), 1019–1040.

Tiano, J. C., Witbaard, R., Bergman, M. J. N., Rijswijk, P. Van, Tramper, A., Oevelen, D. Van, & Soetaert, K. (2019). Acute impacts of bottom trawl gears on benthic metabolism and nutrient cycling. *ICES Journal of Marine Science*. https://doi.org/10.1093/icesjms/fsz027